# Weather-Adaptive Regenerative Braking Strategy Based on Driving Style Recognition for Intelligent Electric Vehicles

**DOI:** 10.3390/s25041175

**Published:** 2025-02-14

**Authors:** Marwa Ziadia, Sousso Kelouwani, Ali Amamou, Kodjo Agbossou

**Affiliations:** 1Department of Mechanical Engineering, Hydrogen Research Institute, University of Québec at Trois-Rivières, Trois-Rivieres, QC G8Z 4M3, Canada; sousso.kelouwani@uqtr.ca (S.K.); ali.amamou@uqtr.ca (A.A.); 2Department of Electrical and Computer Engineering, Hydrogen Research Institute, University of Québec at Trois-Rivières, Trois-Riviere, QC G8Z 4M3, Canada; kodjo.agbossou@uqtr.ca

**Keywords:** intelligent electric vehicle, adaptive regenerative braking, weather conditions, driving style recognition, machine learning, energy recovery optimization

## Abstract

This paper examines the energy efficiency of smart electric vehicles equipped with regenerative braking systems under challenging weather conditions. While Advanced Driver Assistance Systems (ADAS) are primarily designed to enhance driving safety, they often overlook energy efficiency. This study proposes a Weather-Adaptive Regenerative Braking Strategy (WARBS) system, which leverages onboard sensors and data processing capabilities to enhance the energy efficiency of regenerative braking across diverse weather conditions while minimizing unnecessary alerts. To achieve this, we develop driving style recognition models that integrate road conditions, such as weather and road friction, with different driving styles. Next, we propose an adaptive deceleration plan that aims to maximize the conversion of kinetic energy into electrical energy for the vehicle’s battery under varying weather conditions, considering vehicle dynamics and speed constraints. Given that the potential for energy recovery through regenerative braking is diminished on icy and snowy roads compared to dry ones, our approach introduces a driving context recognition system to facilitate effective speed planning. Both simulation and experimental validation indicate that this approach can significantly enhance overall energy efficiency.

## 1. Introduction

Transportation electrification is a pivotal advancement in reducing greenhouse gas emissions (GHG). The International Energy Agency projects that by 2030, the global fleet of electric vehicles (EVs) will reach 245 million, highlighting the shift toward large-scale electrification [1,2]. However, the development of EVs faces challenges, primarily due to the limitations of current battery technology, which is characterized by low energy density. The driving range of EVs depends on the capacity of the battery pack and the efficiency of the powertrain system. Enhancing battery capacity and optimizing powertrain efficiency aims to reduce energy consumption per kilometer, thereby extending the driving range of EVs [3]. Moreover, combining autonomous driving with transportation electrification has led to the integration of Autonomous Electric Vehicles (AEVs). This synergy offers numerous benefits that can further improve the use of EVs, including enhanced safety, automated charging processes, and optimized energy consumption [4,5,6].

A key feature of EVs is their regenerative braking system, which converts kinetic energy back into stored energy in the battery pack [7,8]. In practice, braking forces in EVs are divided between hydraulic braking and regenerative braking. Given the variable braking demands of drivers, it is essential to allocate specific margins for hydraulic braking to ensure a smooth and comfortable braking experience [9,10]. Furthermore, the integration of ADAS can improve safety during intelligent braking and further optimize energy recovery. When the vehicle can anticipate and autonomously respond to upcoming deceleration needs, these systems can significantly maximize energy regeneration [11].

Connected and autonomous vehicles, through Intelligent Transportation Systems (ITS), offer valuable predictive insights into upcoming road conditions. These insights enable the proactive planning of deceleration speed profiles, allowing for smoother and more efficient management of unavoidable deceleration events [12]. Recent advancements in this field focus on anticipating future events to minimize irreversible driving errors across diverse road conditions [13,14]. Additionally, ref. [15] presents an MPC-based approach that emphasizes tire dynamics and vehicle load control, enhancing regeneration efficiency and minimizing tire slip loss. Another study by [16] employs Q-learning algorithms to generate optimal speed profiles during red light stops, utilizing real-time parameters through vehicle-to-infrastructure (V2I) communication. This approach has been experimentally verified to yield improved outcomes by adjusting vehicle braking actions under varying road conditions. Furthermore, ref. [17] explores a hybrid method known as the layer hidden Markov model–dynamic fuzzier neural network, which improves the accuracy in recognizing driver braking intentions. This advancement increases braking sensitivity and boosts the amount of recoverable energy. Overall, the literature demonstrates significant progress in anticipating future events, optimizing the use of regenerative braking, and reducing reliance on hydraulic braking systems [18,19].

The driver’s style significantly influences the effectiveness of regenerative braking [20,21,22], as do the physical constraints of motor generation and the system’s deceleration limits. Several studies in the literature focus on optimizing energy regeneration within these physical constraints [23]. For example, ref. [24] uses real driving test data to map the physical constraints of regenerative braking, which informs the development of a deceleration planning system aimed at optimizing energy consumption and enhancing recuperation efficiency. These optimization strategies are increasingly integrated into ADAS, primarily intended for fully autonomous (Level 5) vehicles. However, their effectiveness may be limited when applied to Level 3 or Level 4 autonomous vehicles, where driver involvement remains crucial. A notable challenge at these levels is ensuring driver acceptance of the braking control strategies, as highlighted in the literature [25]. Incorporating the driver’s naturalistic driving style into these systems is essential. More specifically, integrating individualized regeneration performance into ADAS models and detecting moments when this performance is optimal can reduce the risk of driver frustration caused by inappropriate or poorly timed advice and warnings. This personalization not only enhances the user experience but also plays a pivotal role in increasing the effectiveness of eco-feedback technologies [26]. The evaluation of regenerative braking performance is closely tied to the driver’s ability to regenerate energy up to the engine’s regeneration capacity while adhering to their preferred maximum speed [27]. Over the past two decades, there has been a growing emphasis on developing customized strategies that cater to the unique needs and preferences of individual drivers [28,29]. Drivers often exhibit distinct speed preferences between stopping points, which are critical to consider, as these preferences directly correlate with both the regeneration limit and the individual’s regenerative performance [30]. Integrating these preferences into the regenerative braking model and tailoring the braking control to align with individual energy regeneration capabilities adds an important dimension that enhances user acceptance of optimal braking strategies [31]. The approach introduced in [32] aims to balance optimal braking control with naturalistic regeneration behavior, accommodating individual drivers’ speed preferences and naturalistic energy regeneration performance during trips between two stopping points. The study proposes an Adapted Energy Deceleration Planning Strategy (AEDPS) that requires 30 s power forecasts to estimate naturalistic energy regeneration performance. Based on these predictions, it dynamically adjusts the deceleration planning horizon to optimize energy recovery. While this research was conducted under typical conditions, such as during summer, it is important to recognize that realistic driving scenarios often involve significant variations in weather, which can greatly affect driving dynamics and system efficacy.

Extensive research has established statistical links between driving style in various environmental contexts, such as car-following and lane-keeping, and different weather conditions [33,34]. These studies collectively demonstrate how factors like impaired visibility, precipitation, and extreme temperatures significantly influence driver behavior and vehicle handling. For instance, driving simulation experiments indicate that fog reduces both driving speeds and acceleration rates while increasing the distance drivers maintain from the vehicle ahead [35]. Beyond general environmental influences, specific factors such as road surface friction coefficients under varying weather conditions directly impact energy consumption. Research has shown that different weather conditions significantly affect traffic flow and vehicle operation. For example, a study conducted in the Salt Lake Valley found that commencement delays increase by 5% on wet roads and by 23% on snowy roads compared to dry conditions [36]. Additionally, a novel approach integrating control barrier functions (CBFs) and control Lyapunov functions (CLFs) has been developed to enable adaptive cruise control that adjusts to varying surface conditions affected by weather [37]. Another study utilized the Adams/Car Simulator to investigate how friction coefficients impact the dynamic responses of various vehicle types—sedans, buses, and trucks. The findings revealed significant variations in friction coefficients across different weather conditions, 0.5 for wet, 0.4 for rainy, 0.28 for snowy, and 0.18 for icy conditions, all of which critically affect braking performance. In contrast, higher friction coefficients of 0.9 to 0.6, typical of dry conditions, did not significantly alter braking distances [38]. Overall, these insights underline the importance of considering environmental factors when designing vehicle control systems and optimizing energy usage.

Different classification models are employed to recognize driving styles, utilizing various artificial intelligence algorithms. Studies referenced in [39,40] highlight that the most commonly used algorithms include Fuzzy Logic (FL), Random Forests (RF), k-nearest Neighbor (kNN), Support Vector Machine (SVM), and Long Short-Term Memory (LSTM) networks. Additionally, research in [41] shows that XGBoost offers superior predictive accuracy over RF and SVM models, particularly in contexts involving traffic crash fatalities and the differentiation between safe and unsafe driving styles. Ref. [42] explores using raw stemmatics data for driving style classification, which is pivotal for improving driving risk assessments, fuel efficiency, and supporting adaptive driving assistance and insurance practices. The features used in this analysis include instantaneous, short-term, and long-term styles derived from acceleration and speed measurements. Traditional classification methods such as k-NN, SVM, and Decision Trees are applied to time series data. The results reveal that Decision Trees, when used with the proposed features, surpass RNN-based approaches in classification accuracy, thus proving the effectiveness of these features in conjunction with decision Trees. In parallel, some studies have begun to explore the development of a driving style identifier (DBI) as a novel approach to classifying driving styles. For instance, ref. [43] introduces a novel method designed to regulate torque demand in EVs that are equipped with single-pedal driving (SPD). This method focuses on efficiently interpreting driving styles to promote eco-driving. Central to this approach is the development of a Driving Behavior Identifier (DBI), which utilizes the Binary Dragonfly Algorithm (BDA) combined with an Adaptive Neuro-Fuzzy Inference System enhanced by Particle Swarm Optimization (ANFIS-PSO). Additionally, to further optimize powertrain control, Torque Demand Look-Up Tables (TDLTs) are created using the Whale Optimization Algorithm (WOA). This comprehensive approach aims to refine the responsiveness and efficiency of the vehicle’s powertrain system by adapting torque output based on real-time driving style analysis. The study also underscores the critical role of advanced classification models, such as Decision Trees, SVM, and ANFIS, in recognizing driving styles under varying weather conditions.

Research on regenerative braking primarily focuses on optimizing energy efficiency under specific weather conditions. Some studies target summer conditions, while others explore strategies adapted to icy roads during winter. For instance, refs. [44,45] investigate speed control in low-adhesion contexts. Ref. [44] proposes an energy controller for electric vehicles on sloped roads, incorporating a slip compensation algorithm. This controller, based on a three-degree-of-freedom dynamic model, improves energy consumption even on slippery surfaces while remaining robust to variations in vehicle mass and battery state. Similarly, ref. [45] introduces an autopilot designed for low-friction roads, utilizing a two-level hierarchical optimization to prevent skidding and maintain stable performance on surfaces with variable friction. The deceleration planning also contributes to addressing challenges on low-friction roads. Ref. [46] introduces a coordinated control strategy (eMPC-CCS) for regenerative braking systems (RBSs) and anti-lock braking systems (ABSs) in electric vehicles. This strategy specifically tackles the difficulties associated with low-adhesion road conditions, such as icy or wet surfaces, using a Model Predictive Control framework. The proposed approach enhances real-time coordination of braking forces, ensuring vehicle stability and optimized braking performance even when tire–road adhesion is significantly reduced. It minimizes control conflicts between an RBS and ABS while employing state error compensation to maintain robust performance across varying road conditions. These advancements highlight the critical role of adaptive and predictive strategies in maximizing safety and energy recovery on low-adhesion roads. Similarly, the work of [47] focuses on optimizing energy regeneration in electric vehicles through an Energy-Optimal Adaptive Cruise Control (EACC) strategy based on Model Predictive Control (MPC). This methodology effectively manages low-adhesion conditions by dynamically adjusting model parameters, improving vehicle stability, and minimizing energy losses. However, ref. [47] does not address complex climatic variations, which are essential for practical applications in diverse real-world scenarios. The study’s methodology is limited to optimizing parameters related to road adhesion and does not provide a strategy adapted to frequent transitions between varying weather conditions (e.g., from dry to icy roads). This shortcoming reduces the strategy’s effectiveness in varied weather conditions, where driver-assistance systems may generate unnecessary alerts or fail to achieve optimal energy performance. Our study proposes an innovative approach: an adaptive regenerative braking strategy that integrates the dynamics of varying weather conditions and driving styles. Unlike previous works, our method relies on an advanced driving style recognition model that identifies road conditions (dry or icy) and slope profiles (flat or inclined) while dynamically adjusting the deceleration horizon. This strategy maximizes energy recovery and reduces unnecessary alerts in complex environments, such as icy or sloped roads. The main objective of this research is to develop an energy-efficient and adaptive regenerative braking strategy, with several key contributions: (1) Dynamic adaptation of braking performance to weather conditions and driving styles. (2) Reduction of false alerts, particularly in varied road contexts. (3) Maximization of energy recovery through advanced driving style recognition and optimization algorithms. These contributions address the growing need for braking assist systems that combine adaptability, energy efficiency, and comfort in diverse driving environments.

The remainder of this article is organized as follows. Section 2 provides an overview of driving style and naturalistic energy regeneration performance, comparing winter and summer conditions. Section 3 details the proposed method for the weather-adaptive regenerative braking strategy. Experimental results are reported in Section 4, and finally, Section 5 presents the conclusion.

## 2. Overview of Driver Style and Naturalistic Energy Regeneration Performance: Icy vs. Dry Roads

This section outlines a research methodology to examine variations in drivers’ braking styles across different weather conditions, specifically contrasting the challenges of winter, characterized by snowy weather and icy roads, with the more predictable conditions of summer, which feature dry roads. The focus is on how drivers naturally adjust their acceleration and deceleration in response to changing weather and road conditions and how these adjustments subsequently impact energy regeneration performance.

### 2.1. Reaction to Maximum Speed and Acceleration

This study involves three drivers, each participating in ten driving experiments to ensure a comprehensive analysis. These experiments are evenly divided between five trials conducted during snowstorms on icy roads and five conducted under sunny conditions on dry roads, resulting in a total of thirty driving instances. The experiments take place on a standardized five-kilometer stretch of road featuring seven stop signs, ensuring consistency across all trials. Data are collected on maximum speed, cruising distance, and other critical metrics, such as longitudinal acceleration and jerk. To enhance the analysis of weather-related impacts on driving styles, we combine experimental data with publicly available data from the SHRP 2 Naturalistic Driving Study (NDS) to extend our database. This comprehensive dataset captures real-world driving behavior under various weather conditions, including rain and snow, complementing the experimental controlled data collected in our study. This integration ensures compatibility by harmonizing key variables, such as speed, acceleration, and braking patterns, across both data sources. Using the strengths of both experimental precision and the representativeness of public data, we validate our findings and improve the robustness of this study. Including diverse datasets ensures a more reliable evaluation of the model’s performance across a wide range of environmental scenarios, highlighting its adaptability and effectiveness.

The findings, detailed in Table 1, indicate that drivers adopt a significantly more cautious approach under snowstorm conditions, with notable reductions in speed to enhance safety. Additionally, there is considerable variability in the degree of speed reduction among different drivers, reflecting diverse adaptive strategies in response to icy conditions. This variability provides valuable insights into individual driving styles and their impact on energy regeneration performance in adverse weather conditions. For example, Driver 2, who typically cruises at a maximum speed of 48 km/h on dry roads, shows only a marginal reduction in speed when driving on icy roads, a pattern also observed with Drivers 3 and 4. However, Driver 1, who usually accelerates to 63 km/h in clear conditions, notably decreases their speed to 46 km/h on icy roads. This substantial reduction reflects a significant adjustment, showcasing how Driver 1’s approach to hazardous conditions differs markedly from their usual driving style. Such discrepancies highlight the impact of ingrained driving habits, with drivers like Driver 1, accustomed to higher speeds, needing to make more pronounced speed adjustments in response to adverse weather, unlike their more cautious counterparts. Further analysis identifies the Standard Deviation of Braking Behavior (SDBB) as a crucial metric in evaluating braking performance. A higher SDBB typically indicates a greater likelihood of abrupt braking, a characteristic more prevalent on dry roads where drivers often adopt more aggressive driving styles. Notably, the SDBB for drivers on dry roads is 1.37 times higher than that for drivers in optimal visibility conditions, suggesting a more aggressive braking pattern. These observations are consistent with previous research, which posits that driver style varies significantly between dry and icy conditions, with a noticeable escalation in aggressive actions on dry surfaces.

In summary, the analysis clearly highlights how road conditions influence driving style. Drivers tend to adopt a more cautious approach on icy roads to enhance safety, while dry roads often encourage more aggressive driving and braking styles. This adaptability reflects drivers’ instinctive responses to different environmental conditions. These insights reaffirm the importance of understanding and adjusting to these conditions not only to improve driving safety but also to enhance braking efficiency. This understanding is crucial for developing adaptive driving assistance systems that respond effectively to varying weather and road scenarios.

### 2.2. Reactions to Braking Distance

The speed profile and braking distance data for the three drivers under dry and slippery road conditions are presented as solid lines for dry roads and dashed lines for icy roads, as illustrated in Figure 1. These data demonstrate a significant correlation between weather conditions and braking behavior, highlighting the profound impact of environmental variations on road safety. As weather conditions worsen and road friction decreases, drivers naturally adjust their rates of longitudinal deceleration to ensure safety and comfort in all stopping scenarios. The analysis shows that braking distances on slippery roads are longer compared to those on dry roads, attributed to the lower coefficient of friction on icy surfaces, which necessitates increased caution from drivers. Despite the lower maximum speeds observed on slippery roads, the analysis reveals that high-speed braking can increase braking distances by up to 21.76%. Conversely, during low-speed braking, there is a slight decrease in braking distance of 3.54%. Although this reduction might appear marginal, it highlights the critical need to adapt braking strategies based on both speed and road conditions. These findings underscore the crucial role of weather conditions and road friction in driving style, optimizing enhanced regenerative braking performance.

### 2.3. Energy Regeneration: Dry vs. Icy Roads

Figure 2 presents the power recovery results for different driving styles under varying road conditions, neatly divided into two sections. On the left side, the power of drivers on dry roads is showcased, contrasting with the right side, which depicts conditions on slippery roads. This layout allows for a direct comparison, underscoring that regenerative power on slippery roads is significantly lower than on dry roads due to differences in road friction.

Analyzing the driving styles of two drivers, one navigating icy roads and the other dry ones, reveals distinct patterns in speed profiles and power usage. Observations of sudden fluctuations in these profiles, particularly during short durations, provide valuable insights into how drivers encounter and respond to road irregularities such as gravel, asphalt, or concrete. These fluctuations serve as crucial warning signals, identifying road segments that are likely icy.

Figure 2 effectively demonstrates how high-speed fluctuations correlate with notably low levels of regenerative power, in contrast to segments with stable speed profiles that exhibit higher power values. It is interesting to note that the figure also highlights variable-speed segments that share similar maximum speeds but exhibit significant disparities in output power. The segment outlined by a solid blue triangle and a continuous line shows significantly higher power values. In contrast, the segment marked by a dashed blue triangle displays lower power levels. This difference can be primarily attributed to the influence of the road slope. The first segment (solid blue triangle) corresponds to a nearly flat area, where the vehicle must generate more power to maintain or reach the maximum speed. The second segment (dashed blue triangle) is located on a downhill slope, where inertia and gravity help sustain speed with reduced energy consumption, thereby explaining the lower power values observed.

This analysis clearly shows that road gradient significantly influences regenerative power, along with frictional characteristics and maximum speed, impacting the overall efficiency of the regeneration process. Consequently, any adaptive braking strategy must comprehensively incorporate these parameters to optimize effectiveness and reliability. By accounting for road slope, friction levels, and maximum achievable speed, the adaptive braking system can be fine-tuned to maximize performance across various driving conditions, thereby ensuring both efficiency and safety.

## 3. Weather-Adaptive Regenerative Braking Strategy Design

### 3.1. Control Framework

The proposed approach is structured in two primary stages: driving style recognition and energy-efficient deceleration planning, as illustrated in Figure 3. The first stage focuses on recognizing driving styles using a dataset representing different driving styles across diverse weather conditions, specifically contrasting summer and snowy environments. This dataset includes key parameters of the vehicle, such as speed, torque, acceleration, and power, enabling the development of a driving style recognition model based on a Decision Tree algorithm. Speed constraints are established for each driving style class within this model. In the second stage, the driving style recognition model informs the activation of specific control strategies. For instance, when the model identifies “icy road”, typically associated with snowy conditions, it activates an innovative braking strategy designed to adapt to road friction conditions. This strategy adjusts the deceleration horizon based on the road slope, applying a shorter horizon for downhill slopes while optimizing the deceleration profile, whereas natural driving behavior is considered optimal on flat roads. The resulting optimal control problem is then solved using a dynamic programming algorithm, leveraging the speed constraints defined in the first stage. Alternatively, if the model identifies “dry roads”, typically indicative of summer conditions, it deploys the AEDPS strategy studied in our previous work [32]. AEDPS optimizes deceleration planning by considering individual regeneration performance and driver speed preferences. It requires 30 s power forecasts to predict the force regenerated by the driver, $F_{\mathrm{Rgn}/\mathrm{Nat}}$. Accordingly, the difference between the physical limits of an electrified powertrain, $F_{\mathrm{Rgn}/\mathrm{Lmt}}$, and the force regenerated by the driver, $F_{\mathrm{Rgn}/\mathrm{nat}}$, are expressed as $\Delta F=F_{\mathrm{Rgn}/\mathrm{Lmt}}-F_{\mathrm{Rgn}/\mathrm{nat}}$. This difference, referred to as the energy regeneration maximization rate, is utilized to update the horizon for deceleration planning. Depending on the resulting value of ΔF, the strategy estimates the best deceleration planning horizon—long, medium, or short—to maximize energy recovery during deceleration. The long horizon focuses on enhancing regeneration for a high energy regeneration maximization rate, while the medium and short horizons cater to moderate and minimal recovery needs, respectively. When optimal deceleration styles are identified, the system disables ADAS control warnings to drivers. Additionally, the methodology integrates an Adaptive Neuro-Fuzzy Inference System (ANFIS) classification model to enhance the prediction accuracy of the driver’s naturalistic regeneration behavior to estimate the energy regeneration maximization rate. In this scenario, the optimal control problem is again addressed using dynamic programming, with the speed constraint informed by the driving style recognition established in the first stage.

### 3.2. Fixed Route and Experimental Setup for Data Collection

This study employs the fixed route, which applies both to vehicles in which the driver takes the same route to work and to commercial vehicles, such as city buses and garbage trucks, that mostly commute on the same routes simultaneously. The selection of fixed routes in this study is used to improve the forecasting of variation in driving styles’ on-road characteristics.
There were no surrounding vehicles, traffic lights, or dynamic obstacles.The braking events were initiated by static obstacles, such as stop signs and static objects detected by Mobileye.The route covered a total distance of 4 km.The selection of the road was made mainly because of the substantial number of stop panels (12 stop signs).
All tests were conducted under consistent environmental conditions to ensure the reliability of the results. Wind speed, temperature, and road surface humidity were maintained at constant values throughout the testing process to minimize external variability. This approach ensured that the observed outcomes are attributed solely to the performance of the regenerative braking system and were not influenced by fluctuating environmental factors. An instrumented 2017 Kia Soul intelligent electric vehicle (EV) collected naturalistic driving data, in-vehicle information, and environmental factors. As depicted in Figure 4, the onboard measurement system comprised the global positioning system (GPS), the onboard diagnostic system (OBD), and the ADAS system installed in the experimental vehicle. The GPS system was a Mozaic-X5 (Septentrio, Leuven, Belgium). Figure 4 shows the OBDlinkMx (Phoenix, Arizona, USA) and Mobileye systems connected to the OBD2 ports for CAN-Bus communication (Intel Corporation, Jérusalem, Israël ). The OBDlinkMx operates via Bluetooth on PCs and is responsible for collecting vehicle parameters such as brake and acceleration pedal states, motor torque, etc. Mobileye can recognize obstacles at a longitudinal distance of up to 250 m from the reference point.

### 3.3. Driving Style Recognition Model and Power Forecasting Using Classification Machine Learning Model

This subsection provides an overview of driving style recognition models, which are typically classified based on the features utilized, the classification outputs they generate, and the specific models employed. Additionally, it introduces a new classification-based prediction model designed to forecast energy regenerated under varying road conditions. This model is a key component of the AEDPS, enhancing its ability to adapt to diverse driving environments.

#### 3.3.1. Classification Features

The process of driving style recognition involves extracting and selecting key features from data provided by sensors or devices in vehicles. These features typically include velocity, acceleration, torque, angular velocity, altitude, longitude, and environmental conditions such as weather (coded as snowy = 1, summer = 0) and road slope. While there is no universal set of features that suits every predictive model, acceleration is frequently emphasized in past studies as critical for driving style recognition. Additional factors like jerk, throttle opening, and revolutions per minute (rpm) are also important, particularly for estimating braking smoothness and identifying cautious driving styles on slippery roads. We selected these features based on their demonstrated relevance in previous studies. For instance, ref. [48] highlights their essential role in identifying driving behaviors, whether aggressive or cautious. Similarly, ref. [49] emphasizes their effectiveness in recognizing driving patterns under different road conditions. In our previous research [32], we also explored this issue by applying Principal Component Analysis (PCA) to reduce the dimensionality of the dataset used in the prediction model. The correlations between the principal components and the original variables are presented in Table 2, revealing no correlation between them, thereby ensuring their statistical independence. The first principal component was found to be strongly correlated with six key variables: battery current, battery voltage, battery power, motor torque, and, to a lesser extent, the distance-to-stop and speed scores. These results indicate that these six criteria vary together: an increase in one typically leads to an increase in the others. Furthermore, this component is particularly influenced by battery-related variables. The second principal component, on the other hand, is primarily correlated with rotational speed and its equivalent speed. It also shows a strong correlation with road characteristics obtained via GPS data. These analyses indicate that the following features are discriminative and relevant to our study. They are particularly effective in capturing driver responses to environmental factors, such as icy or wet roads, directly influencing driving style recognition.

#### 3.3.2. Classification Outputs

The outputs of driving style recognition models are categorized into distinct classes, each labeled according to the style it represents. This systematization helps in accurately interpreting the varied styles drivers exhibit on the road. Research in this area has introduced various labeling schemes to capture the nuances of driver style, typically using classifications such as normal vs. aggressive or calm vs. aggressive. Although some studies suggest using four or more labels for a more detailed characterization, such implementations are less common due to the increased complexity and potential challenges in maintaining model accuracy. In this study, however, a more detailed approach is employed, resulting in six specific classes: driver 1 on dry roads, driver 1 on icy roads, driver 2 on dry roads, driver 2 on icy roads, driver 3 on dry roads, and driver 3 on icy roads. These classes are distinctly labeled to differentiate styles under varying weather conditions, such as drivers on dry versus icy roads. This granular categorization provides a deeper insight into how different environmental contexts influence driving styles. Additionally, the model’s capability to categorize many drivers accommodates the diversity and complexity found in real-world driving scenarios. This enhanced adaptability improves the model’s utility and applicability in broader transportation and safety applications, facilitating more effective decision-making based on a comprehensive analysis of driving styles.

#### 3.3.3. Classification Models

The driving style recognition models evaluated in this study include Decision Trees, SVM, and ANFIS, each selected for its unique strengths and proven effectiveness. Decision Trees stand out for their simplicity and high classification accuracy, particularly when applied to time series data. SVM excels in managing non-linear patterns and high-dimensional datasets, ensuring reliable differentiation of driving styles. Meanwhile, ANFIS combines the advantages of Fuzzy Logic and neural networks, allowing for in-depth and nuanced analysis of driving behaviors. These models were chosen based on their complementary strengths and their demonstrated success in prior studies, including research published in [48]. Our research goes beyond a straightforward comparison of Decision Trees, SVM, and ANFIS by focusing on enhancing their accuracy through systematic optimization of structural and training parameters. To achieve this, we employ advanced feature extraction and selection techniques, ensuring that the models are equipped to interpret complex datasets with greater precision. These methodologies are central to improving the robustness and effectiveness of the models, enabling them to capture intricate patterns and deliver reliable driving style recognition across diverse scenarios.

#### 3.3.4. Forecasting Power Model: Icy vs. Dry Roads

Given the significant differences in regeneration behavior between icy and dry roads, and considering the diversity of driving styles under various road conditions captured in the database utilized in this new approach, it is essential to employ a new power prediction model based on classification to enhance forecast accuracy. This is particularly crucial because power prediction plays a key role in braking control when utilizing AEDPS. Consequently, ANFIS is chosen as one of the most advanced and popular models in the current state of the art. Its natural capability to handle complex structures makes it an ideal choice for this application. Notably, ANFIS is based on a Fuzzy Logic classifier specifically designed for range estimation, which is a crucial and timely research topic in the field of electric vehicles today [50].

The ANFIS model presents unique challenges in time series analysis due to its lack of a distinct periodic nature, rendering it prone to neither converging nor diverging. This sensitivity to initial conditions is a benchmark issue in the domains of both neural network and fuzzy modeling. To overcome these challenges, researchers have utilized the fourth-order Runge–Kutta method to derive numerical solutions for the Mackey–Glass equation, which facilitates the extraction of time series values at integer intervals. The ANFIS model is trained to predict time series data defined by the Mackey–Glass (MG) time-delay differential equation:(1)$$\dot{x}\left(t\right)=\frac{0.2x(t-\tau )}{1+x^{10}\left(t-\tau \right)}-0.1x\left(t\right)$$

Assuming initial conditions ***x***(0) = 1.2, τ=17, and x(t)=0 for t<0, the MG time series can be generated. To plot the MG time series, use the provided command. The time series prediction process involves using existing data points up to the point of prediction x=t+P to estimate the value at a future time point.

The conventional approach for such predictions establishes a correspondence between input vector from *D* points spaced Δ apart, specifically (x(t−(D−1)Δ),…,x(t−Δ),x(t)), to predict a future value x(t+P). This method adheres to the customary parameters used to forecast the MG time series. If we assume the input vector *D* = 4 and the predicted future point Δ = *P* = 6,(2)x(t+6)=F[x(t),x(t−6),x(t−12),x(t−18)]

To address the limitations of the ANFIS prediction model, we implemented targeted measures to improve data quality, mitigate biases, and optimize computational efficiency. A diverse and representative dataset was curated that included various driving styles, weather conditions, and road types. Preprocessing, including normalization, outlier removal, and class balancing, ensured robust data quality. Biases were identified and corrected through exploratory data analysis and a systematic evaluation framework. To address computational constraints, we reduced rule complexity using Fuzzy C-Means clustering, utilized parallel computing to accelerate training, and optimized parameters with advanced algorithms. These efforts significantly improved the robustness, efficiency, and real-world applicability of the model.

#### 3.3.5. Validation Model

The effectiveness of our approach, particularly in terms of precision, is evaluated using the following accuracy formula for the driving style recognition model:(3)$$\mathrm{Accuracy}=\frac{TP+TN}{TP+TN+FP+FN}$$

Here, True Positives (TP) and True Negatives (TN) represent correctly predicted positive and negative instances, respectively. False Positives (FP) and False Negatives (FN) represent incorrectly predicted positive and negative instances, respectively.

The main goal of the forecasting power model is to minimize the discrepancy between measured and forecasted regenerative power over the next 12 s. The evaluation of regenerative power forecasting accuracy involves five metrics. Although Mean Absolute Percentage Error (MAPE) is commonly used for forecasting accuracy, it is criticized for producing undefined or extreme values when actuals are zero or close to zero, respectively. Therefore, this study employs alternative metrics that do not suffer from these limitations: R-Squared, Mean Absolute Error (MAE), Normalized Mean Absolute Error (NMAE), Symmetric Mean Absolute Percentage Error (sMAPE), and Root Mean Squared Error (RMSE). The equations for these metrics are detailed in Table 3.

### 3.4. *Energy-Efficient Deceleration Planning*


The energy-efficient deceleration planning algorithm shown in Algorithm 1 is designed to optimize braking based on weather and road conditions. It takes as input a driving style class *C*, which includes speed constraints, and the slope $\theta_{i}$. For each driving style class $c_{i}$ in *C* and slope condition $\theta_{i}$, the algorithm selects the appropriate braking strategy. If $c_{i}$ is classified as “dry”, the algorithm applies the AEDPS strategy and computes an optimal sequence constrained by speed limits suitable for dry conditions. If $c_{i}$ is classified as “icy” and has “downhill” slopes, a shorter planning horizon is used, optimizing the deceleration profile by considering speed constraints that are adapted to icy conditions as defined by the driving style class *C*. For “icy” and “flat” conditions, the algorithm adopts a naturalistic driving behavior with optimal driving parameters and disables alerts. The output is an optimal deceleration profile $a^{*}$ and a deceleration planning horizon *h*. The algorithm for deceleration planning is presented below.
**Algorithm 1** Energy-Efficient Deceleration Planning**Require:** 
driving styles class *C* (includes speed constraints), Slope θ1:**for** each slope condition $\theta_{i}$ **do**2:    **if** $c_{i}=\mathrm{dry}$ **then**3:        return←
AEDPS strategy4:        Set Optimal sequence $\{a^{*}\}\leftarrow$
Result BASED ON AEDPS5:    **else if** $c_{i}=\mathrm{icy}\;\mathrm{and}\;\theta_{i}=\mathrm{downhill}$ **then**6:        Planning horizon←short7:        Set Optimal sequence $\{a^{*}\}\leftarrow$
Result based on icy road friction8:    **else if** $c_{i}=\mathrm{icy}\;\mathrm{and}\;\theta_{i}=\mathrm{flat}$ **then**9:        return←
Naturalistic driving behavior10:    **end if**11:**end for**12:**return** 
$\mathrm{Optimal}\;\mathrm{deceleration}\;\mathrm{profile}\;\{a^{*}\},Planninghorizonh$

#### 3.4.1. Model of Braking Deceleration

The braking deceleration dynamics are modeled to determine the most effective deceleration pattern. This method leverages the fundamental functionality of an Anti-lock Braking System (ABS) to calculate the optimal braking deceleration [51]. The slip ratio, denoted by *s*, is a crucial metric in this analysis, quantifying the extent of sliding between the tire and the road surface. It is mathematically defined as(4)$$s=\frac{\omega \cdot r-v}{\max(\omega \cdot r,v)}$$
where ω represents the rotational speed of the wheel, *r* denotes the wheel radius, and *v* signifies the longitudinal velocity of the wheel, which is commonly approximated by the vehicle speed in practical scenarios.

Furthermore, the braking force coefficient $\varphi_{b}$, which is distinct from the friction coefficient, varies based on the type of resistive force involved and is defined as the ratio of the brake force exerted on the ground to the perpendicular load. This coefficient can be found in the literature or determined experimentally. It is formally expressed as(5)$$\varphi_{b}=\frac{F_{X}}{F_{Z}}$$
The braking force coefficient $\varphi_{b}$ varies as a function of the slip ratio. Figure 5 illustrates the dynamic relationship between $\varphi_{b}$ and the slip ratio, emphasizing how changes in the slip ratio influence braking performance, particularly on icy surfaces. Initially, in phase OA, the slip ratio is low, indicating no slippage between the tire and the road. As the slip ratio increases and transitions into phase AB, the braking force coefficient $\varphi_{b}$ rises progressively under favorable road conditions, eventually reaching an optimal level at point *B*. Beyond this point, in phase BC, the slip ratio increases, causing the wheel to skid completely on the surface. This skidding drastically reduces the braking force coefficient, $\varphi_{b}$, to its lowest level, severely diminishing braking efficiency.

In reference to the slip ratio illustrated in Figure 5, point *O* represents the origin coordinate, while point $B(s_{p},\varphi_{p})$ denotes the peak coefficient of friction. Point *A* has coordinates $\left(K_{1}\cdot s_{p},K_{2}\cdot \varphi_{p}\right)$, where the inequality $\frac{\varphi_{p}-K_{2}\cdot \varphi_{p}}{s_{p}-K_{1}\cdot s_{p}}<\frac{K_{2}\cdot \varphi_{p}}{K_{1}\cdot s_{p}}$ holds. Additionally, point *C*$\left(s_{s},\varphi_{s}\right)$ represents the sliding coefficient of friction, with the slip ratio reaching 100%.

During phase OA, the slip ratio remains relatively low. Despite the presence of a slip ratio due to changes in the rolling radius, actual sliding between the tire and the ground does not occur. Upon application of the braking force, the contact area ahead of the tire tread extends slightly, with the rate of increase in the rolling radius being directly proportional to the braking force. As a result, the curve of $\varphi_{b}$ can be approximated to a straight line, expressed by the assumed linear relationship(6)$$\varphi_{b}=a_{1}s$$

If the coordinates of point A are obtained, the slope $a_{1}$ can be calculated using Equation (Equation 6):(7)$$a_{1}=\frac{K_{2}\cdot \varphi_{p}}{K_{1}\cdot s_{p}}$$

During phase AB, the slip ratio reaches its optimal state due to favorable road conditions. Within this phase, localized sliding occurs between the tire and the ground, resulting in a deceleration of the growth rate of $\varphi_{b}$ until the optimal slip rate is achieved at point *B*. The curve representing $\varphi_{b}$ is approximated as a quadratic curve within this phase. The equation for the curve is given by(8)$$\varphi_{b}=a_{2}s^{2}+bs$$

If the coordinates of points *A* and *B* are known, the coefficients $a_{2}$ and *b* can be calculated using the following equations:(9)$$\left\{\begin{array}{c} a_{2}=\frac{K_{1}^{2}-K_{2}}{2s_{p}\left(K_{1}-K_{1}^{2}\right)}\\ b=\frac{\left(K_{2}-K_{1}^{2}\right)\varphi_{p}}{\left(K_{1}-K_{1}^{2}\right)s_{p}} \end{array}\right.$$

In phase BC, the higher the slip rate, the smaller the braking force coefficient $\varphi_{b}$ becomes, reaching 100% at point *C*, indicating complete slippage of the wheel on the ground. In this phase, the curve of $\varphi_{b}$ can be approximated by a linear relationship, assumed as(10)$$\varphi_{b}=a_{3}s+c$$

If the coordinates of points *B* and *A* are known, the coefficients $a_{3}$ and *c* can be calculated using the equations(11)$$\left\{\begin{array}{c} a_{3}=\frac{\varphi_{s}-\varphi_{p}}{1-s_{p}}\\ c=\frac{\varphi_{p}-\varphi_{s}\varphi_{p}}{1-s_{p}} \end{array}\right.$$

According to Equation (Equation 4), the maximum braking deceleration is given by(12)$$a_{b\;\max}=\varphi_{b}g$$
where *g* is the acceleration due to gravity.

Two brake parameters, pressure-regulating frequency and adjusting amplitude, are incorporated into the model, affecting the braking deceleration as follows:(13)$$a=a_{b\;\max}-\frac{1}{2}\delta +\frac{1}{2}\delta \cos\omega \left(t\right)$$

Here, ω represents the regulating frequency, which varies from 19rad/s to 126rad/s depending on vehicle speed and road condition, and δ is the adjusting amplitude, calculated as(14)δ=(φ(0.2)−φ(0.15))·g

Assuming the road surface is fine, and substituting Equation (Equation 7) for $\varphi_{b}$, the braking deceleration can be modeled as(15)$$a=g\left(a_{2}s^{2}+bs\right)-\frac{\delta }{2}+\frac{\delta }{2}\cos\omega \left(t\right)$$

#### 3.4.2. Energy Optimization on Icy Roads by Dynamic Programming

In the proposed strategy, the dynamic programming (DP) method is employed to optimize the deceleration profile. This method takes into account the environmental conditions of EV operation, including the slope and deceleration constraint of driving style on icy roads.

The braking force in vehicle dynamic models can be mathematically expressed as follows:(16)$$\begin{array}{c} F_{\mathrm{Brk}}\left(t\right)=F_{\mathrm{Rgn}}\left(t\right)+F_{\mathrm{Frc}}\left(t\right)=Ma\left(t\right)+\frac{1}{2}\rho AC_{d}v{\left(t\right)}^{2}+Mg\left(\mu \cos\left(\theta \right)+\sin\left(\theta \right)\right) \end{array}$$

The vehicle’s speed, mass, acceleration, cross-sectional area, and frontal drag coefficient, are represented by v(t), *m*, a(t), *A*, and $C_{d}$, respectively. $F_{\mathrm{Frc}}\leq 0$ is the frictional force, and $F_{\mathrm{Rgn}}\leq 0$ is the energy-regenerative deceleration force. Assume that $F_{\mathrm{Frc}}\approx 0$.

The speed profile is a temporal and spatial process, which makes v(X(t)) represent the speed profile at a specific observation station (i.e., location) at time *t*. Furthermore, the speed profile and deceleration duration can be described as follows:(17)$$\begin{array}{cc} t_{N} & =\sum_{i=0}^{N-1}\frac{\Delta X_{i}}{v_{i}} \end{array}$$

Assume the deceleration starts at vehicle position $X_{k}$ and finishes at position $X_{N}$. For any position $X_{i}$ where k≤i≤N, the slope $\theta_{i}$ and the vehicle’s longitudinal speed $v_{i}$ and acceleration $a_{i}$ at $X_{i}$ are known. The acceleration formulas under different road conditions, defined by $a_{i/\mathrm{dry}}$ on dry roads, and $a_{i/\mathrm{icy}}$ on icy roads, are given by(18)$$a_{i}=\left\{\begin{array}{c} a_{i/\mathrm{dry}}=g\left(a_{2}s^{2}+bs\right)-\frac{\delta }{2}+\frac{\delta }{2}\cos\omega \left(t\right)\\ a_{i/\mathrm{icy}}=g\left(a_{3}s+bs\right)-\frac{\delta }{2}+\frac{\delta }{2}\cos\omega \left(t\right) \end{array}\right.$$

The total energy $E_{N}$ during the vehicle’s deceleration on icy roads from $X_{k}$ to $X_{N}$ is expressed as(19)$$E_{N}=E_{k}+\sum_{i=k+1}^{N}F_{\mathrm{Rgn}}\left(v_{i/\mathrm{icy}},a_{i/\mathrm{icy}},\theta_{i}\right)\Delta \dot{X}$$
where $E_{k}$ is the kinetic energy at $X_{k}$ and $F_{\mathrm{Rgn}}\left(v_{i/\mathrm{icy}},a_{i/\mathrm{icy}},\theta_{i}\right)$ < 0 represents the regenerative force profile from $X_{k}$ to $X_{N}$. $\Delta \dot{X}={\dot{X}}_{i+1}-{\dot{X}}_{i}$.

The optimal energy during the deceleration on icy roads is calculated as(20)$$E_{N}^{*}=\min_{a_{i}}\left[E_{k}+\sum_{i=k+1}^{N}F_{\mathrm{Rgn}}\left(v_{i/\mathrm{icy}},a_{i/\mathrm{icy}},\theta_{i}\right)\Delta \dot{X}\right]$$(21)$$a_{\min}\leq a_{i/\mathrm{icy}}\leq a_{\max}$$

Using DP, the optimal sequence $\left\{a_{i/\mathrm{icy}}^{*}\right\}=\left\{a_{i/\mathrm{icy}},k+1\leq i\leq N\right\}$ is determined by(22)$$\left\{a_{i/\mathrm{icy}}^{*}\right\}=\arg\min_{a_{i}}\left[E_{k}+\sum_{i=k+1}^{N}F_{\mathrm{Rgn}}\left(v_{i/\mathrm{icy}},a_{i/\mathrm{icy}},\theta_{i}\right)\Delta \dot{X}\right]$$

Given $a_{i/\mathrm{icy}}^{*}$, the optimal deceleration speed profile is obtained by integrating $\left\{a_{i/\mathrm{icy}}^{*}\right\}$. Subsequently, the objective function is defined based on the predicted braking style from the recognition model, incorporating road and weather condition data. Constraints are then established reflecting the specific driving style on road conditions. An optimization algorithm is subsequently applied to determine the optimal deceleration profile, taking into account the defined objective function and the set constraints specific to icy roads. Under conditions of permanent ice, the regeneration effect is determined by integrating the value of $a_{i/\mathrm{icy}}^{*}$ into the control model using dynamic equations linked to the slip ratio. A deceleration profile, dynamically adjusted based on weather conditions and slope, allows the regenerative braking to be proactively modulated to prevent a significant decrease in efficiency caused by excessive slippage. This procedure facilitates dynamic adjustment of braking strategies, enhancing vehicle safety and performance across various road and weather conditions.

## 4. Results and Discussion

This section assesses the benefits of the proposed WARBS strategy. The evaluation begins with a simulation to assess the performance of the driving style recognition model using MATLAB R2023a (MathWorks, Natick, MA, USA) Next, the assessment of energy regeneration efficiency enhancement on mixed icy and dry roads is conducted.

### 4.1. Results of Driving Style Recognition Model

#### Classifier Model Comparison

Figure 6, Figure 7 and Figure 8 present the classification results for three different models: Decision Tree, ANFIS, and SVM. Despite the similarity in power regeneration amplitudes across different drivers, our proposed method demonstrates exemplary performance. The Decision Tree model, in particular, achieves 100% accuracy during both the validation and testing phases. For all classification strategies, the dataset was split into 67% for training and 33% for testing. The confusion matrices for each model are displayed in the corresponding figures. The classes are numbered as follows:-Class 1: Driver 1 on icy roads (Dr1/Icy)-Class 2: Driver 1 on dry roads (Dr1/Dry)-Class 3: Driver 2 on icy roads (Dr2/Icy)-Class 4: Driver 2 on dry roads (Dr2/Dry)-Class 5: Driver 3 on icy roads (Dr3/Icy)-Class 6: Driver 3 on dry roads (Dr3/Dry)

The drivers are analyzed based on distinct characteristics, such as maximum speed, longitudinal acceleration, and jerk values (rate of change of acceleration). Driver 1 exhibits a dynamic driving style, particularly noticeable on dry roads, with higher maximum speeds and acceleration values than the other drivers. On icy roads, while speeds are reduced, the style remains relatively aggressive, with significant variations in acceleration and jerk parameters. Driver 2 adopts a moderate style, characterized by lower maximum speeds and more stable driving. The smooth speed adjustments and minimal variations in acceleration values reflect a cautious approach, especially in slippery conditions. Driver 3 stands out with a conservative driving style, featuring significantly lower maximum speeds and reduced jerk and acceleration values, demonstrating an adaptation to slippery road conditions. Table 4 shows an RMSE of 0.038 for SVM, 0.0071 for ANFIS, and 0 for Decision Tree. For the SVM model, presented in Figure 8, the most common misclassification involves samples from Class 1 (Dr1/Icy) being incorrectly classified as Class 3 (Dr2/Icy), with six misclassified samples, constituting only 0.0013% of all samples from Class 1. The ANFIS, shown in Figure 6, also misclassifies samples from Class 1 as Class 3 at a rate of 2.24% and samples from Class 5 (Dr3/Icy) as Class 3 at a rate of 7.34%. The Decision Tree model, illustrated in Figure 7, achieves the highest overall accuracy of 100%, making it particularly effective in identifying the six specified driving styles, thereby validating its application in our strategy. The analysis of the results shows that driving characteristics such as speed, acceleration, and jerk are essential for distinguishing driving styles under varying conditions. The differences observed between Drivers 1, 2, and 3 highlight the models’ ability to effectively recognize behaviors based on weather conditions. These findings further support the relevance of the models and their suitability for the proposed strategy.

### 4.2. Results of Power Forecasting-Based Classification Method

To evaluate the performance of the proposed ANFIS-based classification forecasting method for power regeneration, we present the results in Figure 9. These results are compared with those from the Long Short-Term Memory (LSTM) forecasting method proposed by [32], validating the necessity, reliability, and effectiveness of our approach. The dataset utilized for forecasting consists of driving data from three drivers across two weather conditions, summer and snowy, resulting in six distinct driving style variations, with each behavior documented over ten trips. Figure 9 illustrates the comparison of precision rates between the ANFIS model and the LSTM model, using RMSE as a metric for accuracy in power regeneration prediction. The LSTM model records an accuracy of 0.63 kW, a decrease from the 0.4402 kW reported in [32] for summer styles alone, attributed to the inclusion of icy road data. In contrast, the ANFIS model shows significant improvement, achieving an RMSE of 0.3677 kW. Table 5 details the performance of the ANFIS model with various membership functions, highlighting that the psigmf membership function yields the highest precision. Typically, classification-based prediction models outperform non-classification models, as rule-aligned consumption models exhibit greater regularity and predictability compared to general consumption models. This higher precision is expected, as the complexity inherent in non-classification approaches tends to reduce prediction accuracy. In conclusion, the proposed method successfully classifies time series data, achieving a high level of prediction accuracy. Furthermore, the expanded dataset facilitated the generation of more profiles through classification, which contributed to reduced volatility and enhanced accuracy in load predictions.

### 4.3. Enhancing Energy Regeneration Efficiency on Mixed Icy and Dry Roads


The Table 6 assess the performance of the proposed WARBS, drivers’ naturalistic regeneration performance on mixed and dry roads, as well as the AEDPS designed specifically for dry roads, were provided for comparison. To facilitate this comparison, tests were conducted on a route composed of five distinct segments, each designed to offer varied conditions. The road friction levels and slopes differed between these segments, allowing for a comprehensive evaluation of the braking efficiency of each strategy.

The results provides detailed driving parameters for three drivers (Driver 1, Driver 2, and Driver 3) across five segments. The table includes information on slope types, regeneration forces under three scenarios (naturalistic driving, AEDPS strategy, and WARBS approach) and the corresponding deceleration planning horizons. Specifically, the numbers listed in the table indicate the performance for Driver 1, Driver 2, and Driver 3 in each row under their respective scenarios. Importantly, all three approaches were tested by the same drivers, ensuring consistency in driving behavior and conditions across the different scenarios. The regeneration forces obtained under the AEDPS strategy $F_{\mathrm{Rgn}/\mathrm{Opt}}$ are calculated using an electric vehicle model designed to maximize power regeneration up to the powertrain limit while optimizing vehicle speed. Similarly, the regeneration forces from the WARBS approach are optimized for vehicle speed while adapting to varying weather conditions. For comparison, the regeneration forces from naturalistic driving $F_{\mathrm{Rgn}/\mathrm{Nat}}$ are estimated based on drivers’ behaviors under mixed road conditions (dry and icy). The table highlights significant variability in regeneration forces across segments and drivers. For example, Driver 1 exhibits $F_{\mathrm{Rgn}/\mathrm{Nat}}$ values ranging from −1559 N to −3745 N in segment 1, reflecting differences in driving behavior. Generally, regeneration forces are lower on icy segments (segments 2, 4, and 5) compared to dry segments, indicating the limitations of naturalistic driving under low-traction conditions. Additionally, the AEDPS and WARBS strategies consistently outperform naturalistic driving in terms of power regeneration, with optimized regeneration forces closely tailored to slope and weather conditions.

The AEDPS strategy generally improves energy regeneration compared to naturalistic driving, although the level of improvement varies by segment. For instance, for Driver 1 in segment 3, $F_{\mathrm{Rgn}/\mathrm{Opt}}=-2210$ N (AEDPS) compared to −384 N (naturalistic driving). Similarly, the results demonstrate that AEDPS effectively adjusts the deceleration planning horizons based on naturalistic regeneration performance on dry roads. For example, in segment 1, Driver 2 is evaluated as performing optimally, and AEDPS offers no significant improvement in regeneration. This is primarily because the driver’s naturalistic regeneration performance is close to the powertrain’s regeneration limit. However, on icy road segments, the AEDPS strategy leads to false alerts and suboptimal performance, as it is not designed to handle road conditions other than dry surfaces. This limitation underscores the need for a more versatile approach capable of addressing the complexities of mixed road conditions. These findings reveal that while AEDPS provides gains over naturalistic driving, it demonstrates clear limitations under extreme conditions, particularly on icy roads. The table also highlights that only the WARBS strategy is capable of maximizing regeneration under all road conditions, whether icy or dry, by planning deceleration with a short horizon on downhill slopes. This demonstrates that WARBS significantly outperforms AEDPS across most segments, especially in icy conditions. For instance, for Driver 3 in segment 4, the optimal regeneration force achieved with WARBS is $F_{\mathrm{Rgn}/\mathrm{Opt}}$ = −893 N compared to only −549 N with AEDPS, showcasing a substantial improvement. These results confirm that the braking assistance provided by the WARBS strategy excels in both efficiency and adaptability, offering a high rate of user acceptance due to its reliable alerts and scalability, which effectively address diverse weather conditions.

Figure 10 and Figure 11 illustrate the maximization of regeneration on segments with icy and dry roads, respectively. These figures present the distance traveled, speed, planning horizon, and profiles of both natural and optimized regeneration forces for electric vehicles. As shown in Figure 10a, the driver’s speed profile on flat icy roads generates a maximum natural regeneration force, $F_{\mathrm{Rgn}/\mathrm{Nat}}$ (blue lines), of −450 N, which is considered optimal. In contrast, Figure 10b displays the driver’s speed profile on downhill slopes, suggesting a short-term deceleration profile that improves regeneration from −140 N to −450 N (green line). Furthermore, the force profile demonstrates that significant deceleration on a downhill road results in substantial transient regeneration forces. However, this is not the case for cruising modes, which lead to low demands for transient regeneration forces. This analysis highlights the importance of adaptive deceleration planning to maximize energy recovery, particularly under varying road slope conditions. Figure 11 illustrates the results when the driving style is classified by the model as occurring on dry roads. In this scenario, the strategy adopts a long-term planning horizon due to a significant difference, $\Delta F=F_{\mathrm{Rgn}/\mathrm{Lmt}}-F_{\mathrm{Rgn}/\mathrm{Nat}}$, as previously discussed. Additionally, $F_{\mathrm{Rgn}/\mathrm{Opt}}$ (green lines) reaches the regeneration capacity limit of the powertrain, $F_{\mathrm{Rgn}/\mathrm{Lmt}}$ (red lines), maximizing energy recovery without losses caused by slippage. Similar to Figure 10, the convergence trend of $F_{\mathrm{Rgn}/\mathrm{Opt}}$ on road slopes demonstrates that the strategy effectively accounts for road load variations and confirms its ability to track the regeneration limit. This validates the robustness of the approach in maximizing energy recovery under dry road conditions.

Table 7 compares the performance of each method relative to naturalistic driving. The percentage improvement in performance (PI) is calculated as follows:(23)$$PI\left(\%\right)=\frac{F_{\mathrm{method}}-F_{\mathrm{Rgn}/\mathrm{nat}}}{F_{\mathrm{Rgn}/\mathrm{nat}}}\times 100$$
where

-PI: Performance improvement (percentage of performance improvement).-$F_{\mathrm{method}}$: Regeneration force obtained by the method being evaluated (e.g., AEDPS or WARBS).

The results from scenarios on icy roads 2, 4, and 5 indicate that efficiency improvement rises from zero with the application of AEDPS to significant levels with the implementation of WARBS, varying according to driving style and segment. Moreover, both methods exhibited consistent performance in dry segments. Consequently, the proposed method achieved optimal regeneration efficiency under both dry and icy road conditions, a distinction not matched by the standard AEDPS method. This conclusion is further supported by Table 8, which compares the efficiency of both methods and demonstrates the added value of WARBS. Notably, compared to AEDPS, WARBS achieved energy regeneration maximization improvement for all trajectories of 11.6% for Driver 1, 10.4% for Driver 2, and 10.31% for Driver 3, highlighting its effectiveness not only on dry roads but also on icy roads.

The results highlight the critical importance of adopting an adaptive braking strategy, enabling significant maximization of energy regeneration under mixed conditions, including both dry and icy roads. The approach proposed in this study overcomes the limitations of other braking strategies, which are typically confined to specific weather conditions, either exclusively dry or icy.

**Table 7 sensors-25-01175-t007:** The percentage of performance improvement of the methods compared to naturalistic driving.

		Driving Segment				
Method	Driver	1	2	3	4	5
PI_*AEDPS*_ (%)	1	124.3	0	475.5	0	0
	2	0	0	7.04	0	0
	3	23.73	0	21.45	0	0
PI_*WARBS*_ (%)	1	124.3	47.91	475.5	54.14	44.30
	2	0	29.10	7.04	20	181.81
	3	23.75	51.40	21.45	62.65	40.83

**Table 8 sensors-25-01175-t008:** Comparison of AEDPS and WARBS.

	Driving Segment				
Driver	1	2	3	4	5
1	0	47.91	0	54.14	44.30
2	0	29.10	0	20	181.81
3	0	51.40	0	62.65	40.83

## 5. Conclusions

This research makes several noteworthy contributions to the field of regenerative braking. First, the driving style recognition model effectively differentiates between behaviors on icy and dry roads, enabling the implementation of customized braking strategies tailored to varying conditions. Second, dynamic programming adjusts deceleration horizons in real time, striking an optimal balance between safety and energy recovery. Finally, simulations and experimental validations confirm the system’s ability to maximize energy recovery under challenging conditions, such as icy roads and steep slopes. From a practical perspective, this methodology optimizes energy recovery by fine-tuning braking forces in low-adhesion scenarios, enhancing safety through adaptive control, and demonstrating its feasibility for real-world applications. Notably, this strategy achieved significant energy recovery improvements, with regeneration gains of 11.6% for Driver 1, 10.4% for Driver 2, and 10.31% for Driver 3, calculated as averages across all tested trajectories. These results underline its effectiveness in optimizing braking performance on both dry and icy roads. Additionally, the approach highlights its adaptability and superior performance to traditional methods, particularly in minimizing false alerts and maximizing deceleration efficiency. The findings offer actionable recommendations for designers and engineers. Adaptive braking systems should prioritize dynamic adjustments to varying road and weather conditions, leveraging advanced sensors for real-time monitoring. In regions with frequent icy conditions, such as northern Europe or mountainous areas, these strategies deliver significant benefits in improving both safety and energy recovery. In contrast, for more stable climates, simplified braking systems can provide cost-effective solutions while maintaining essential functionality. Furthermore, dynamic programming and predictive models should be explored to refine braking systems, tailoring them to specific driving behaviors and environmental scenarios. While this study represents a significant step forward in regenerative braking strategies, future research could further enhance the model by incorporating additional environmental factors. For example, integrating wind speed as a dynamic parameter could offer deeper insights into its effects on energy recovery and vehicle stability. Wind resistance, particularly at higher speeds, significantly impacts energy consumption, and real-time wind speed estimation could help optimize braking strategies to mitigate these effects. Additionally, accounting for the influence of ambient temperature on battery performance and energy regeneration efficiency would provide a more comprehensive model. Since batteries exhibit reduced efficiency in extreme cold or heat, temperature-dependent adjustments to regeneration settings could maximize energy recovery while protecting battery health. In conclusion, this study offers an adaptive, efficient, and practical solution to regenerative braking, tailored to diverse climatic and road conditions. It provides valuable insights for the development of next-generation electric vehicles, advancing energy optimization, safety, and system reliability in a variety of driving scenarios.

## Figures and Tables

**Figure 1 sensors-25-01175-f001:**
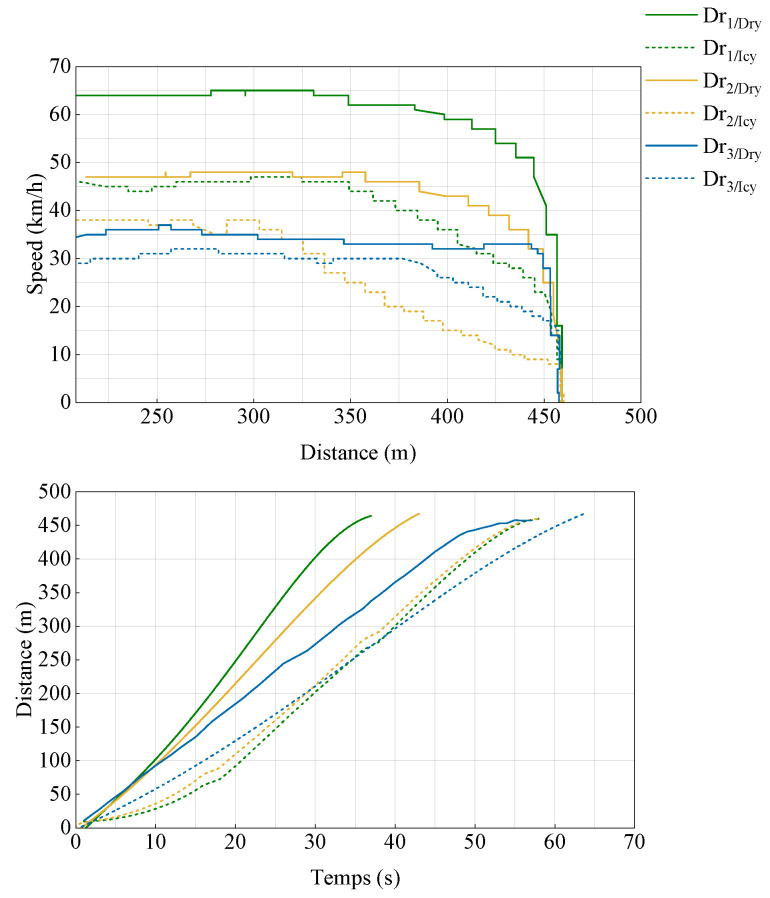
Reactions to braking distance.

**Figure 2 sensors-25-01175-f002:**
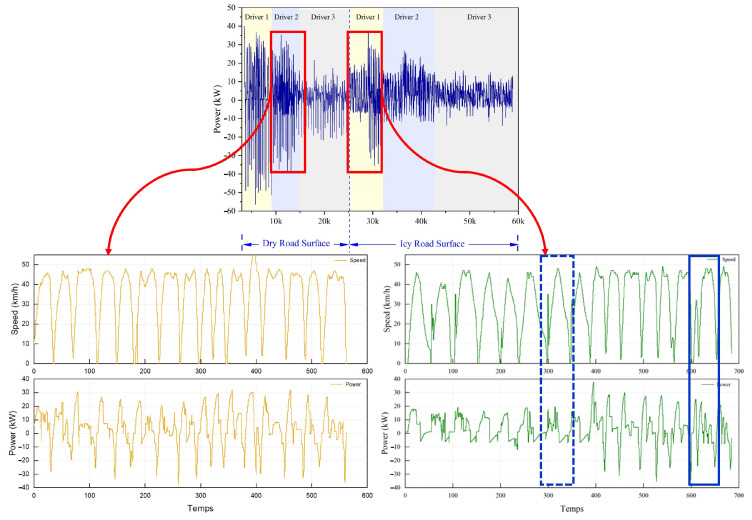
Reactions to power regeneration.

**Figure 3 sensors-25-01175-f003:**
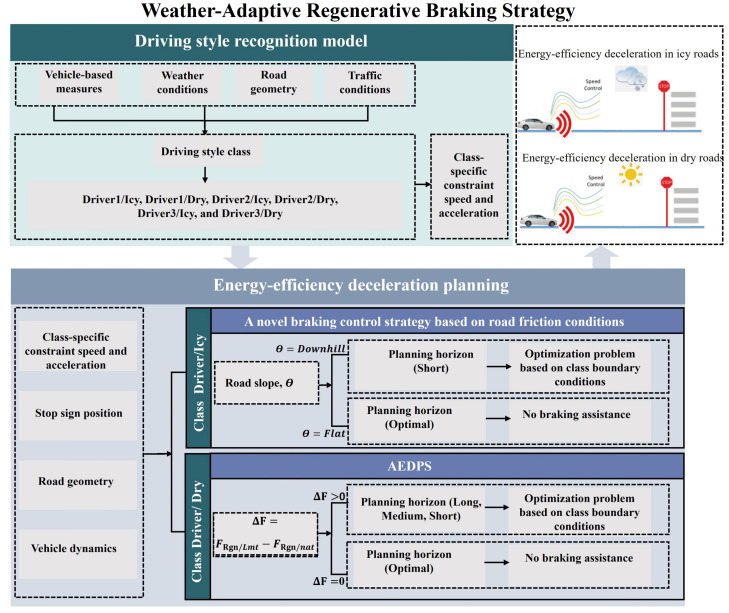
Weather-Adaptive Regeneration Braking Strategy Design.

**Figure 4 sensors-25-01175-f004:**
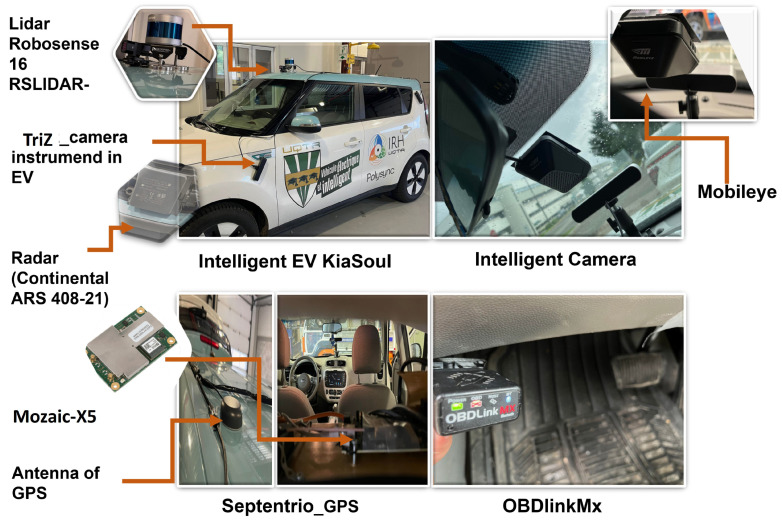
Instrumented intelligent electric vehicle (Kia Soul 2017).

**Figure 5 sensors-25-01175-f005:**
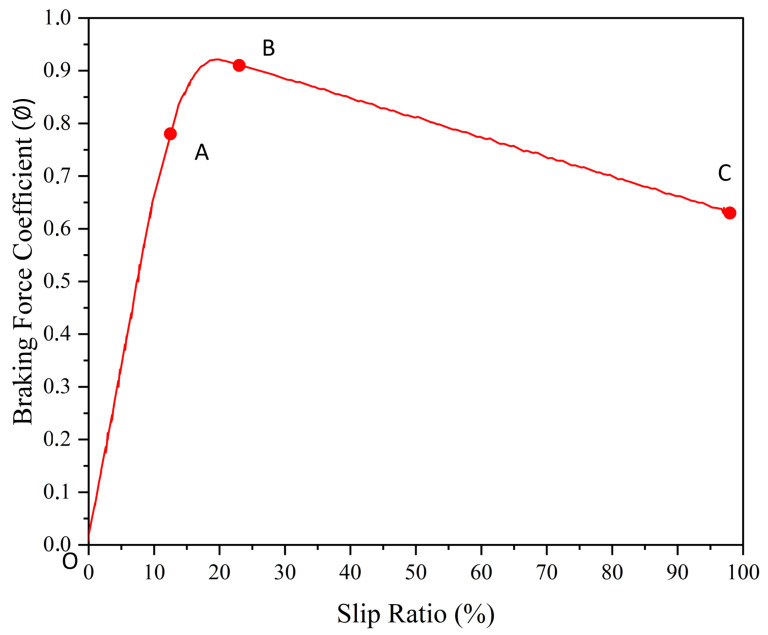
Road friction [51].

**Figure 6 sensors-25-01175-f006:**
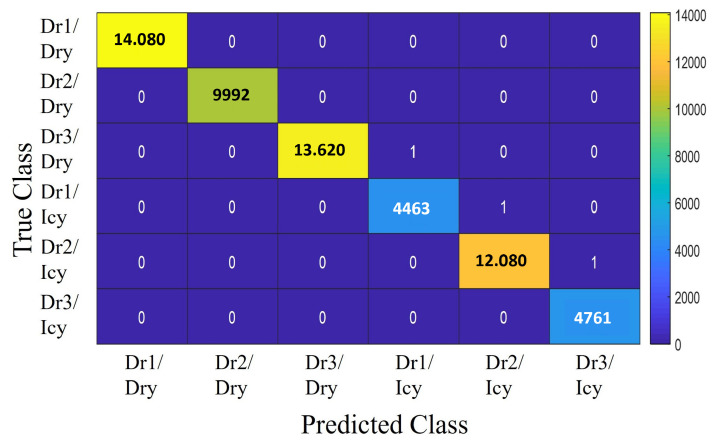
Confusion matrix of ANFIS.

**Figure 7 sensors-25-01175-f007:**
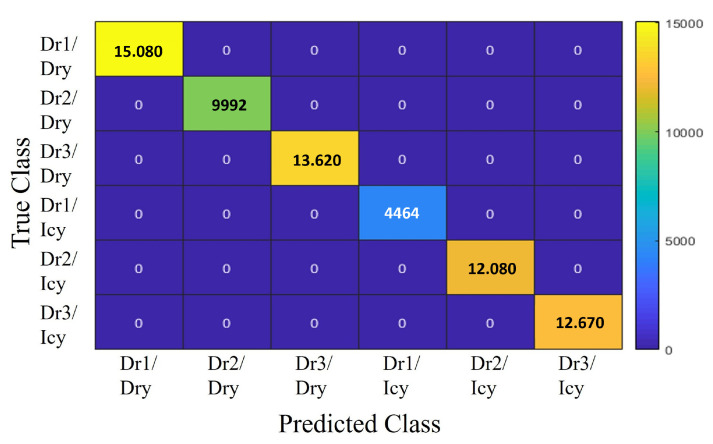
Confusion matrix of Decision Tree.

**Figure 8 sensors-25-01175-f008:**
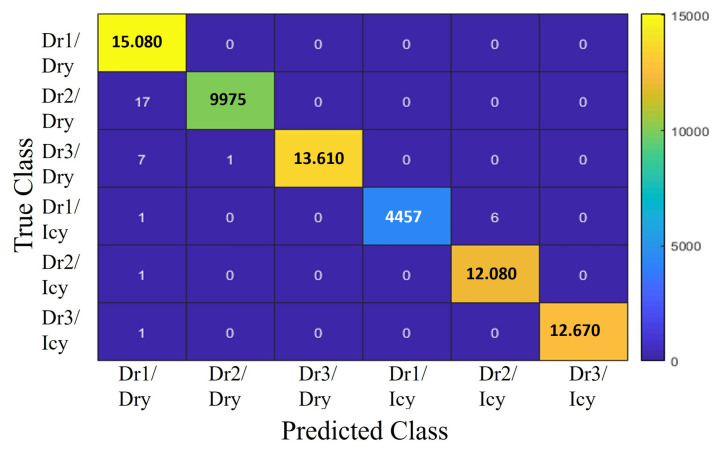
Confusion matrix of SVM.

**Figure 9 sensors-25-01175-f009:**
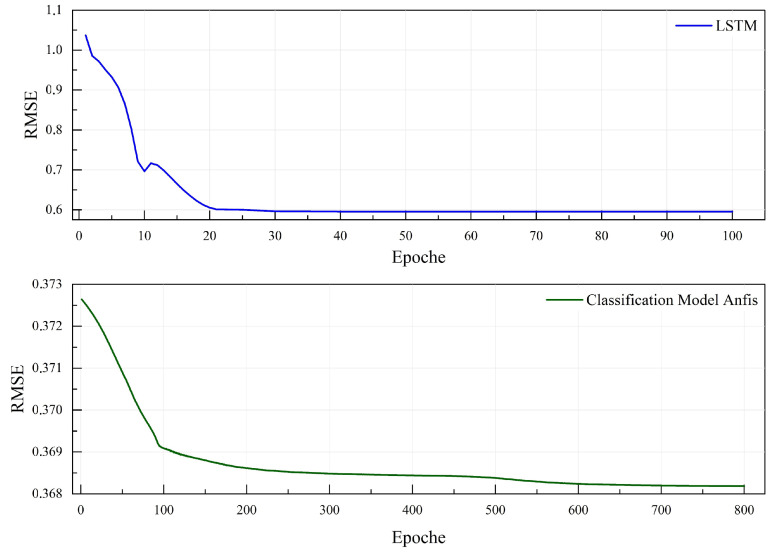
Comparing the RMSE of the proposed ANFIS model with that of the LSTM methodology, excluding classification.

**Figure 10 sensors-25-01175-f010:**
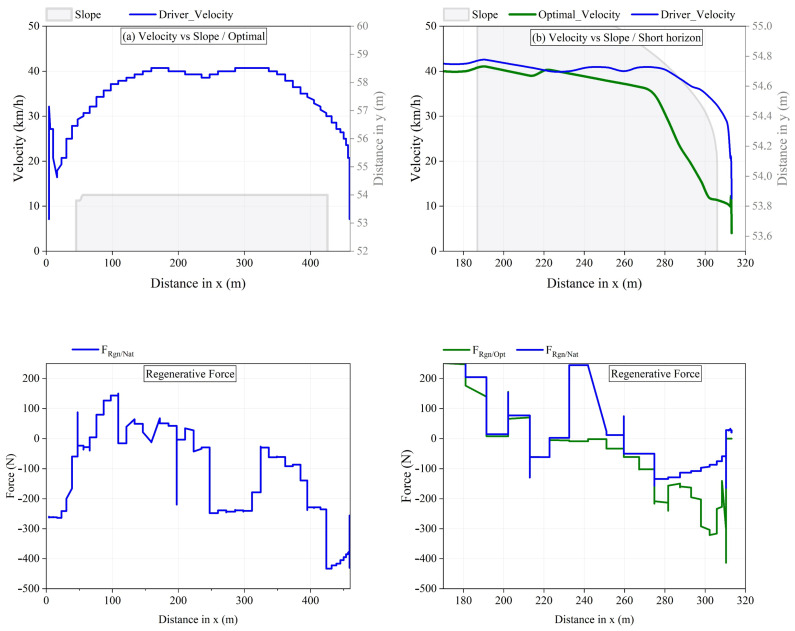
Regeneration maximization on icy segments by WARBS.

**Figure 11 sensors-25-01175-f011:**
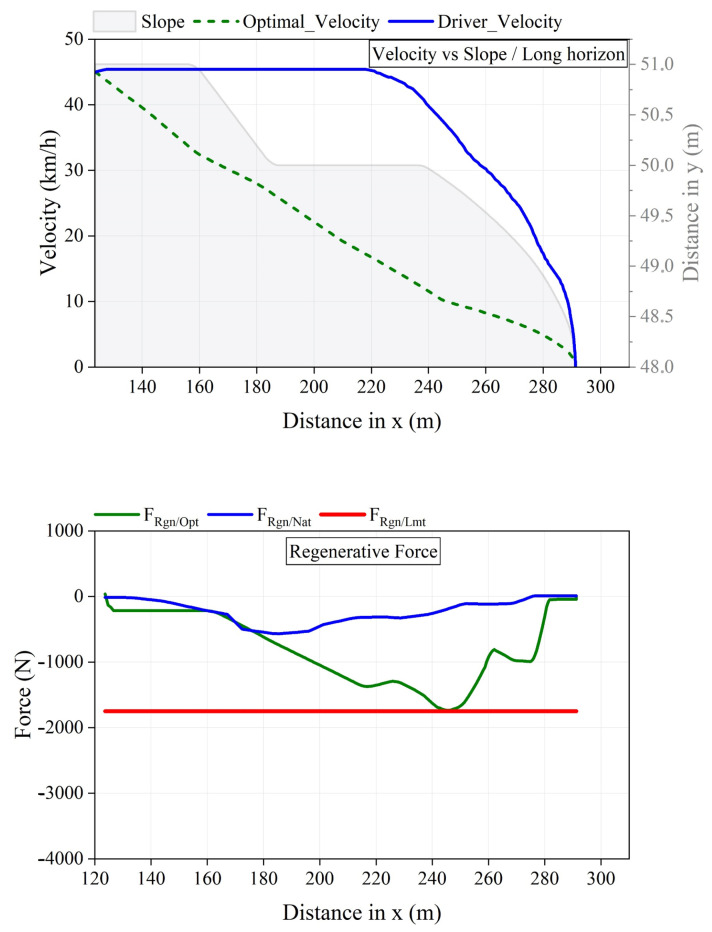
Regeneration maximization on dry segments by WARBS.

**Table 1 sensors-25-01175-t001:** Analysis of driving styles.

Parameters	Drivers	Dry Roads			Icy Roads		
		Max	Mean	SDBB	Max	Mean	SDBB
Speed (km/h)	$DR_{1}$	63	34.85	20.78	46	28.00	13.79
	$DR_{2}$	48	33.81	15.00	42	22.23	13.56
	$DR_{3}$	36	26.68	11.25	30	23.57	8.69
Longitudinal acceleration (m$/s^{2}$)	$DR_{1}$	3.234	−0.003	0.38	2.646	−0.072	0.216
	$DR_{2}$	3.03	−0.0005	0.34	1.862	−0.0501	0.241
	$DR_{3}$	2.84	−0.07	0.22	1.47	−0.084	0.191
Jerk ($m/s^{3}$)	$DR_{1}$	26.75	0.027	3.75	16.85	0.64	2.13
	$DR_{2}$	35.18	0.004	3.29	19.6	0.451	2.44
	$DR_{3}$	16.46	0.63	2.307	18.81	0.765	1.919

**Table 2 sensors-25-01175-t002:** Extracted vectors.

Data	Coefficients of PC1	Coefficients of PC2
Speed	0.23125	0.32684
Distance to stop	0.32548	0.19156
Longitude	0.07202	−0.56701
Latitude	0.06528	0.29689
Altitude	−0.09792	0.54841
Battery current	0.45743	0.01558
Battery voltage	−0.4649	−0.02672
Battery power	0.45818	0.01615
Rotational speed	0.17716	0.33945
Motor torque	0.39247	−0.07497
Acceleration	−0.02749	−0.15498

**Table 3 sensors-25-01175-t003:** Evaluation performance model metric.

Performance Metric	Definition
Mean Absolute Error (MAE)	$MAE=\frac{1}{N}\sum_{i=1}^{N_{i}}\left|Py_{i}-Y_{j}\right|$
Mean Relative Absolute Error (MRAE)	MRAE $=\frac{1}{N}\sum_{i=1}^{N_{i}}\frac{\left|Y_{i}-P_{y_{i}}\right|}{\left|y_{i}\right|}$
Root Mean Square Error (RMSE)	RMSE $=\sqrt{\frac{1}{N}\sum_{i=1}^{N_{i}}{\left(Py_{i}-Y_{i}\right)}^{2}}$
Coefficient of Determination $R^{2}$	$R^{2}=1-\frac{\sum_{i=1}^{i}{\left(Y_{i}-P_{y}\right)}^{2}}{\sum_{i=1}^{n}{\left(Y_{i}-{\bar{Y}}_{i}\right)}^{2}}$

**Table 4 sensors-25-01175-t004:** Result of classification model.

Algo	RMSE	MAE	$R^{2}$
Decision Tree	0	0	1
SVM^multiclass^	0.0388	0.000073	0.9996
ANFIS	0.0071307	0.0000508	1

**Table 5 sensors-25-01175-t005:** Anfis result of forecasting power with different membership functions.

Membership Function	MAE	RMSE	$R^{2}$
trimf	0.942	0.38069	0.9670
gaussmf	1.0213	0.3701	0.9639
gauss2mf	1.0548	0.36841	0.9629
gbellmf	1.0445	0.3688	0.9631
dsigmf	1.0402	0.36773	0.9618
psigmf	1.04027	0.3677	0.9618
pimf	1.0649	0.3681	0.9628

**Table 6 sensors-25-01175-t006:** Driving segment parameters.

		Driving Segment				
Parameter	Driver	1	2	3	4	5
Slope	All	Downhill/dry	Downhill/icy	Flat/dry	Downhill/icy	Downhill/icy
FRgn/nat (N) from naturalist driving	1	−1559	−649	−384	−495	−562
	2	−1743	−457	−1263	−350	−110
	3	−3745	−784	−2750	−549	−698
FRgn/opt (N) obtained by AEDPS	1	−3498	−649	−2210	−495	−562
	2	−1743	−457	−1352	−350	−110
	3	−4634	−784	−3340	−549	−698
AEDPS horizon	1	Long	False alert	Long	False alert	False alert
	2	Optimal	False alert	Short	False alert	False alert
	3	Medium	False alert	Long	False alert	False alert
FRgn/opt (N) obtained by WARBS	1	−3498	−960	−2210	−763	−811
	2	−1743	−590	−1352	−420	−310
	3	−4634	−1187	−3340	−893	−983
WARBS horizon	1	Long	Short	Long	Short	Short
	2	Optimal	Short	Short	Short	Short
	3	Medium	Short	Long	Short	Short

## Data Availability

Data are contained within the article.

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
