# Peer review of "Weather-Adaptive Regenerative Braking Strategy Based on Driving Style Recognition for Intelligent Electric Vehicles"

_sensors, 2025, doi:10.3390/s25041175_

Round 1
Reviewer 1 Report
Comments and Suggestions for Authors
The paper was written clearly and soundly with all the appropriate elements, with and extensive introduction section and solid conclusion. There are several minor issues with the paper which should be addressed by authors:
- figure 1 is missing from the document
- text can benefit from a thorough read through, with some minor typos detected
- figure 3 is in more details explained later on in the text and could possibly be introduced later while mentioned at the beginning
- the section below Fig. 9 should also be checked for consistency in naming (Driver 1 is mentioned where it is not clear from the table whether the numbers listed in the text are for the same or different drivers)
Generally the document presents clear methodology and no major issues found.
One final remark, from someone coming from relatively dry climate and the presented improvements in icy conditions would be a benefit in few days a year, could the authors relate the findings to different world climates, stating where would their methodology be of most significance and where it would not make a great difference (and would such distinction make an influence in car manufacturer strategy for product placement)?
Author Response
Comments 1: [figure 1 is missing from the document] |
|
Response 1: Thank you for highlighting this oversight. We agree with your comment. Figure 1 was indeed missing from the original submission. We have now included Figure 1 in the revised manuscript. The figure has been added on Page [7], in the section "[2.2. Reactions to braking distance]", immediately following Line [286]. This addition provides the necessary context and visual support for the discussion in this section. The updated figure has also been labeled and referenced appropriately in the text to ensure clarity. Please see the highlighted changes in the revised version for your review.]”
|
Comments 2: [text can benefit from a thorough read through, with some minor typos detected.] |
|
Response 2: Thank you for pointing this out. We agree with your comment and have conducted a thorough review of the manuscript to address the minor typos and improve the overall readability of the text. Corrections have been made throughout the document to rectify typographical errors and ensure clarity. For your reference: Line 10: (taking into account: considering), Line 18: (efforts to reduce: reducing), Line 36: (In situations where: When the vehicle), Line 55: (The effectiveness of regenerative braking is significantly influenced by the driver’s style [20] [ 21 ] [ 22 ], as well as by the physical constraints of motor generation and the system's deceleration limits.: The driver’s style significantly influences the effectiveness of regenerative braking [20] [ 21 ] [ 22 ], as well as the physical constraints of motor generation and the system’s deceleration limits.), Line 193: (aimed at examining: to examine), Line 203: (take place: occur), Line 267: (of drivers: of drivers), Line 267: (of drivers: of drivers), Line 275: (segments of the road: road segments), Line 300: (vehicle such: vehicle, such), Line 315: (powertrain ,: powertrain,), Line 317: (difference,: difference), Line 319: (; long: ; long), Line 321: (high: a high), Line 330: (is applicable: applies), Line 335: (and dynamic: or dynamic).]”
|
Comments 3: [figure 3 is in more detail explained later in the text and could possibly be introduced later while mentioned at the beginning.] |
|
Response 3: Thank you for pointing this out. I agree with your suggestion that Figure 3, which is explained in greater detail later in the text, could be introduced at a later stage while still being mentioned earlier for context. To address this, I have made the following adjustments: 1. Relocation of Figure 3: Figure 3 has been moved to Section [3.4.1. Model of braking deceleration] Page 14, Line 505], where its details are discussed in depth, to ensure a more logical flow and alignment with the surrounding text.
2. Early Mention of Figure 3: A reference to Figure 3 has been added in Section [2.3. Energy regeneration: dry vs icy roads/Page 6, Line 277], providing a brief context to introduce its relevance early in the manuscript without disrupting the narrative.]”
|
Comments 4: [the section below Fig. 9 should also be checked for consistency in naming (Driver 1 is mentioned where it is not clear from the table whether the numbers listed in the text are for the same or different drivers)] |
|
Response 4: Thank you for highlighting this point. We agree with your observation and have made the necessary changes to ensure consistency in the designation of drivers in the section below Figure 9. To address this, I have made the following adjustments:
The text has been clarified to explicitly indicate that the results mentioned correspond to the performances of Drivers 1, 2, and 3, as presented in Table 5. Each row in the table reflects the specific results for these drivers across the different scenarios (naturalistic driving, AEDPS strategy, and WARBS approach). The updates also specify that all three approaches were tested by the same drivers to ensure consistency in driving behavior and conditions across the various scenarios. These revisions can be found on page [20], in paragraph [2], and line [660]. The revised text now clearly indicates the correspondence between the drivers and the data presented in the table, ensuring improved clarity and consistency. These changes have been marked in red in the revised manuscript for your convenience.]” “[The results for each segment of the route are summarized in Table 5, which provides detailed driving parameters for three drivers (Driver 1, Driver 2, and Driver 3) across five segments. The table includes information on slope types, regeneration forces under three scenarios (naturalistic driving, AEDPS strategy, and WARBS approach), and the corresponding deceleration planning horizons. Importantly, all three approaches were tested by the same drivers, ensuring consistency in driving behavior and conditions across the different scenarios. The regeneration forces obtained under the AEDPS strategy FRgn/Opt are calculated using an electric vehicle model designed to maximize power regeneration up to the powertrain limit while optimizing vehicle speed. Similarly, the regeneration forces from the WARBS approach are optimized for vehicle speed while adapting to varying weather conditions. For comparison, the regeneration forces from naturalistic driving FRgn/Nat are estimated based on drivers’ behaviors under mixed road conditions (dry and icy). The table highlights significant variability in regeneration forces across segments and drivers. For example, Driver 1 exhibits FRgn/Nat values ranging from −1559N to −3745N in segment 1, reflecting differences in driving behavior. Generally, regeneration forces are lower on icy segments (segments 2, 4, and 5) compared to dry segments, indicating the limitations of naturalistic driving under low-traction conditions. Additionally, the AEDPS and WARBS strategies consistently outperform naturalistic driving in terms of power regeneration, with optimized regeneration forces closely tailored to slope and weather conditions.]”
|
Comments 5: [One final remark, from someone coming from relatively dry climate and the presented improvements in icy conditions would be a benefit in few days a year, could the authors relate the findings to different world climates, stating where would their methodology be of most significance and where it would not make a great difference (and would such distinction make an influence in car manufacturer strategy for product placement)?] |
|
Response 5: Thank you for pointing this out. We agree with your comment and appreciate the suggestion to consider the applicability of the findings across different world climates. Therefore, we have added a discussion in the manuscript to address this point. To address this, I have made the following adjustments: We have included a paragraph in Section [Concluson, Page 23, Line 747] discussing how the presented methodology would be particularly significant in regions with prolonged icy or snowy conditions, such as northern Europe, Canada, or mountainous areas where icy roads are frequent. Additionally, in relatively dry or temperate climates, where icy conditions occur only a few days per year, the methodology may have limited direct application but could still provide benefits in terms of overall vehicle safety and energy efficiency during rare adverse weather events. Finally, we have addressed the implications for car manufacturers by noting that the distinction in climate applicability could influence their product placement strategies. For example, in markets with frequent harsh winters, technologies like WARBS could be emphasized to address local driver needs, while in more stable climates, manufacturers might focus on simpler and cost-effective solutions. These additions are highlighted in red in the revised manuscript for your convenience.]” “[In terms of global applicability, the methodology is particularly relevant in regions with frequent icy conditions, such as northern Europe, Canada, and mountainous areas. However, in climates where icy roads are rare, such as arid or tropical regions, the methodology’s impact may be limited to occasional weather events. This distinction could guide car manufacturers in tailoring their product strategies, focusing on markets where icy road safety and energy recovery improvements are most valued, while adapting to less demanding markets with cost-effective alternatives.]”
|
Reviewer 2 Report
Comments and Suggestions for Authors
The paper focuses on a weather-adaptive regenerative braking strategy based on driving style recognition for intelligent electric vehicles. Considering that the energy recovery potential through regenerative braking is reduced on icy and snowy roads compared to dry roads, the approach introduces a driving context recognition system to facilitate efficient speed planning. Both modeling and experimental validation show that this approach can significantly improve overall energy efficiency
However, there are comments on the paper:
1. Experimental or modeling results from other authors should be cited to confirm the adequacy of vehicle regeneration performance.
2. Keywords should be corrected by adding special terms characterizing the research.
3. In the introduction section it is necessary to indicate the novelty of the conducted research. At the end of the introduction section, the purpose of the scientific research should be defined.
4. Figure 1 is missing in the manuscript, it should be cited.
5. AEDPS optimizes energy recovery in power actuators by determining the optimal regeneration force that minimizes the difference between the maximum and natural regeneration force. How is the regeneration effect determined for permanent ice?
6. Some shortcomings of the ANFIS prediction model for distinguishing between different driver behaviors: dependence on the quality and quantity of training data. The model is sensitive to biases and inaccuracies in data representation. The need for significant computational resources to run ANFIS, which can be a challenge in resource-constrained environments. How are these issues addressed in the paper?
7. In addition to wind speed, there are other external factors that affect traffic patterns, e.g. external temperature, roadway moisture. How are these accounted for in the recovery model?
8. More up-to-date publications on electric vehicle energy should be cited in the list of cited sources.
9. The conclusions should be structured, highlighting the main scientific and especially practical results obtained, as well as recommendations for designers and mechanical engineers.
Author Response
Comments 1: [Experimental or modeling results from other authors should be cited to confirm the adequacy of vehicle regeneration performance.]
|
Response 1: Thank you for pointing this out. We agree with your comment and have incorporated citations from other relevant studies to confirm the adequacy of the vehicle regeneration performance discussed in this manuscript. [Therefore, we have cited the work of [Pan et al., 2022 [1]], which presents a novel energy-optimal adaptive cruise control (EACC) strategy for electric vehicles, leveraging model predictive control (MPC) with a focus on energy efficiency. This research is particularly relevant to scenarios involving low-adhesion surfaces, as it integrates predictive control methods to optimize motor torque and balance energy consumption with safety and tracking performance. By incorporating dynamic adjustments to the weight parameters in MPC, the study ensures effective handling of varying road conditions, including low-adhesion environments. This adaptability enhances vehicle stability during deceleration and braking phases, minimizing energy losses while maintaining safe inter-vehicle distances. Furthermore, the proposed approach demonstrates improved energy recovery and braking efficiency compared to traditional ACC strategies, making it a valuable reference for addressing challenges related to icy or slippery roads. These additions have been included in Section [Introduction], Page [4], Paragraph [1], Line [162], and the references have been added to the bibliography. Also, we have cited the work of [Chu et al., 2024 [2]], which introduces a coordinated control strategy (eMPC-CCS) for regenerative braking systems (RBS) and anti-lock braking systems (ABS) in electric vehicles. It particularly addresses the challenges of low-adhesion road conditions, such as icy or wet surfaces, by leveraging a model predictive control framework. The proposed strategy enhances the real-time coordination of braking forces, ensuring stability and optimized braking performance even under conditions where tire-road adhesion is significantly reduced. This approach minimizes control conflicts between the RBS and ABS and employs state error compensation to maintain robust performance across varying road conditions. Such advancements underscore the importance of adaptive and predictive strategies in maximizing safety and energy recovery on low-adhesion roads. These additions have been included in Section [Introduction], Page [4], Paragraph [1], Line [153], and the references have been added to the bibliography.] [1] C. Pan, A. Huang, J. Wang, L. Chen, J. Liang, W. Zhou, L. Wang, J. Yang, Energy-optimal adaptive cruise control strategy for electric vehicles based on model predictive control, Energy 241 (2022) 122793. [2] L. Chu, J. Li, Z. Guo, Z. Jiang, S. Li, W. Du, Y. Wang, C. Guo, RBS and ABS Coordinated Control Strategy Based on Explicit Model Predictive Control, Sensors (Basel, Switzerland) 24(10) (2024) 3076. |
Comments 2: [Keywords should be corrected by adding special terms characterizing the research.] |
Response 2: Thank you for pointing this out. We agree with your comment and have updated the keywords to better reflect the specific terms characterizing the research. The revised keywords now include terms that highlight the methodology, application, and unique aspects of the study. We have added keywords such as "adaptive regenerative braking," " Driving style recognition, "dynamic programming" and "energy recovery optimization" to ensure that the research is more accurately indexed and easier to find for relevant audiences. These updates can be found in the Keywords section, on Page [1], Line [15].]
|
Comments 3: [In the introduction section it is necessary to indicate the novelty of the conducted research. At the end of the introduction section, the purpose of the scientific research should be defined.]
|
|
Response 3: Thank you for pointing this out. We agree with your comment and have revised the introduction section to explicitly highlight the novelty of the research and clearly define the purpose of the study. We have added a paragraph in the Introduction section (Page [4], Paragraph [1], Line [166]) that emphasizes the innovative aspects of the conducted research, particularly focusing on the adaptive deceleration strategies and their application in various weathers conditions]”
“[Research on regenerative braking primarily focuses on optimizing energy efficiency under specific weather conditions. Some studies target summer conditions, while others explore strategies adapted to icy roads during winter. For instance, [ 37] and [ 38] investigate speed control in low-adhesion contexts. [37] proposes an energy controller for electric vehicles on sloped roads, incorporating a slip compensation algorithm. This controller, based on a three-degree-of-freedom dynamic model, improves energy consumption even on slippery surfaces while remaining robust to variations in vehicle mass and battery state. Similarly, [38] introduces an autopilot designed for low-friction roads, utilizing a two-level hierarchical optimization to prevent skidding and maintain stable performance on surfaces with variable friction. The deceleration planning also contributes to addressing challenges on low-friction roads. [39] introduces a coordinated control strategy (eMPC-CCS) for regenerative braking systems (RBS) and anti-lock braking systems (ABS) in electric vehicles. This strategy specifically tackles the difficulties associated with low-adhesion road conditions, such as icy or wet surfaces, by utilizing a model predictive control framework. The proposed approach enhances real-time coordination of braking forces, ensuring vehicle stability and optimized braking performance even when tire-road adhesion is significantly reduced. It minimizes control conflicts between RBS and ABS while employing state error compensation to maintain robust performance across varying road conditions. These advancements highlight the critical role of adaptive and predictive strategies in maximizing safety and energy recovery on low-adhesion roads. Similarly, the work of [40] focuses on optimizing energy regeneration in electric vehicles through an Energy- Optimal Adaptive Cruise Control (EACC) strategy based on Model Predictive Control (MPC). This methodology effectively manages low-adhesion conditions by dynamically adjusting model parameters, improving vehicle stability, and minimizing energy losses. However, [40] study does not address complex climatic variations, which are essential for practical applications in diverse real-world scenarios. Their methodology is limited to optimizing parameters related to road adhesion and does not provide a strategy adapted to frequent transitions between varying weather conditions (e.g., from dry to icy roads). This shortcoming reduces their effectiveness in varied weather conditions, where driver-assistance systems may generate unnecessary alerts or fail to achieve optimal energy performance. Our study proposes an innovative approach: an adaptive regenerative braking strategy that integrates the dynamics of varying weather conditions. Unlike previous works, our method relies on an advanced driving style recognition model that identifies road conditions (dry or icy) and slope profiles (flat or inclined) while dynamically adjusting the deceleration horizon. This strategy maximizes energy recovery and reduces unnecessary alerts in complex environments, such as icy or sloped roads. The main objective of this research is to develop an energy-efficient and adaptive regenerative braking strategy, with several key contributions: (1) Dynamic adaptation of braking performance to weather conditions and driving styles. (2) Reduction of false alerts, particularly in varied road contexts. (3) Maximization of energy recovery through advanced driving style recognition and optimization algorithms. These contributions address the growing need for braking assist systems that combine adaptability, energy efficiency, and comfort in diverse driving environments.]”
|
Comments 4: [Figure 1 is missing in the manuscript; it should be cited]
|
|
Response 4: Thank you for highlighting this oversight. We agree with your comment. Figure 1 was indeed missing from the original submission. We have now included Figure 1 in the revised manuscript. The figure has been added on Page [7], in the section "[2.2. Reactions to braking distance]", immediately following Line [286]. This addition provides the necessary context and visual support for the discussion in this section. The updated figure has also been labeled and referenced appropriately in the text to ensure clarity. Please see the highlighted changes in the revised version for your review]”
|
Comments 5: [AEDPS optimizes energy recovery in power actuators by determining the optimal regeneration force that minimizes the difference between the maximum and natural regeneration force. How is the regeneration effect determined for permanent ice.]
|
|
Response 5: Thank you for pointing this out. We agree with your comment and have clarified how the regeneration effect is determined for permanent ice conditions in the revised manuscript. We have expanded the discussion in Section [3.4.2. Energy optimization in icy roads by dynamic programming], Page [17], Paragraph [1], Line [579], to explain the methodology for determining the regeneration effect on surfaces with permanent ice. The text now explains that under conditions of permanent ice, the regeneration efficiency is determined by incorporating the value of the optimal acceleration in icy road into the control model, using dynamic equations tied to the slip ratio. A deceleration profile, dynamically adapted to weather conditions and road slopes, enables proactive modulation of regenerative braking to mitigate efficiency losses caused by excessive wheel slippage. This approach allows for real-time adjustments to braking strategies, significantly improving vehicle safety and performance across diverse road and weather conditions.]”
|
Comments 6: [Some shortcomings of the ANFIS prediction model for distinguishing between different driver behaviors: dependence on the quality and quantity of training data. The model is sensitive to biases and inaccuracies in data representation. The need for significant computational resources to run ANFIS can be a challenge in resource-constrained environments. How are these issues addressed in the paper?]
|
|
Response 6: Thank you for pointing this out. We agree with your observation and have clarified in the revised manuscript how the identified shortcomings of the ANFIS prediction model were addressed. 1. Quality and Quantity of Training Data: To ensure robust performance, we utilized a diverse and representative dataset collected under various conditions, including driving styles, weather conditions, and road types. Before training the model, data preprocessing steps were performed, including:
 Normalization to standardize variable scales,  Outlier removal to eliminate errors caused by extreme values,  Class balancing to ensure an equitable distribution of categories.
2. Sensitivity to Biases and Inaccuracies:
An in-depth exploratory data analysis phase was conducted to identify and address potential biases in the data, including examining the distributions of features related to driver behavior. We also implemented a data evaluation framework focused on:
 Completeness: Ensuring that the dataset includes representative samples of the target population,  Accuracy: Reducing noise in the data and improving its reliability. These efforts minimized biases and enhanced data quality, improving model generalizability.
3. Computational Resource Requirements: To address the constraints of resource-limited environments, we optimized the ANFIS model by reducing its complexity. This was achieved through:
 Reducing the number of generated rules using clustering techniques such as Fuzzy C-Means to lower data dimensionality and computational overhead,  Utilizing parallel computing platforms during training to accelerate processes while reducing resource demands,  Parameter optimization uses efficient algorithms to balance performance and resource usage.
A detailed discussion of these approaches has been added to the revised manuscript in Section [3.3.4. Forecasting power model: icy vs dry roads], Page [12], Paragraph [5], Line [453], for further clarity.]”
“[To address the limitations of the ANFIS prediction model, we implemented targeted measures to improve data quality, mitigate biases, and optimize computational efficiency. A diverse and representative dataset was curated, incorporating various driving styles, weather conditions, and road types. Preprocessing, including normalization, outlier removal, and class balancing, ensured robust data quality. Biases were identified and corrected through exploratory data analysis and a systematic evaluation framework. To address computational constraints, we reduced rule complexity using Fuzzy C-Means clustering, utilized parallel computing to accelerate training, and optimized parameters with advanced algorithms. These efforts significantly enhanced the robustness, efficiency, and real-world applicability of the model.]”
|
Comments 7: [In addition to wind speed, other external factors affect traffic patterns, e.g., external temperature, and roadway moisture. How are these accounted for in the recovery model?]
|
|
1. Response 7: Thank you for pointing this out. We agree with your comment and appreciate the opportunity to clarify this aspect. Therefore, we have updated the manuscript to address this point explicitly. We have added a clarification in Section [3.2. Fixed route And experimental setup for data collection 329], Page [8], Line [341] to explain that all tests were conducted under consistent weather conditions, including wind speed, temperature, and road surface humidity. This ensures that the influence of external factors on the recovery model was minimized, allowing the analysis to focus solely on the performance of the regenerative braking system under controlled conditions.]”
“[All tests were conducted under consistent weather conditions to ensure the reliability of the results. Wind speed, temperature, and road surface humidity were maintained constant throughout the testing process to minimize external variability. This approach ensures that the observed outcomes are attributed solely to the performance of the regenerative braking system and not influenced by fluctuating weather factors.]” |
Comments 8: [More up-to-date publications on electric vehicle energy should be cited in the list of cited sources?]
|
|
Response 8: Thank you for pointing this out. We agree with your comment and have updated the manuscript to include more recent and relevant publications on electric vehicle energy. These additional citations provide up-to-date context and support for the research. We have added new references from recent publications (2022–2024) focusing on advancements in energy optimization for electric vehicles. These citations have been included in Section [Introduction], Page [2], Paragraph [1], Line [54], where relevant discussions on energy recovery and optimization occur.]”
“[Updated References:
[3] H.M. Hussein, and all, Electric Vehicle Performance Enhancement Utilizing Hybrid Energy Storage Systems, IEEE Vehicle Power and Propulsion Conference, 2024. [4] J. Hong. and all, Energy-Saving Driving Assistance System Integrated With Predictive Cruise Control for Electric Vehicle, IEEE Transactions on Intelligent Vehicles, 2024.]” |
Comments 9: [The conclusions should be structured, highlighting the main scientific and especially practical results obtained, as well as recommendations for designers and mechanical engineers?]
|
|
Response 9: Thank you for pointing this out. We agree with your comment and have revised the conclusion section to provide a more structured presentation of the main scientific and practical results, along with clear recommendations for designers and mechanical engineers. These revisions can be found in Section [Conclusion], Page [23 et 24], Lines [747], and are marked in red in the revised manuscript for clarity.]”
“[This research makes several noteworthy contributions to the field of regenerative braking. First, the driving style recognition model effectively differentiates between behaviors on icy and dry roads, enabling the implementation of customized braking strategies tailored to varying conditions. Second, dynamic programming adjusts deceleration horizons in real time, striking an optimal balance between safety and energy recovery. Finally, simulations and experimental validations confirm the system’s ability to maximize energy recovery under challenging conditions, such as icy roads and steep slopes. From a practical perspective, this methodology optimizes energy recovery by fine-tuning braking forces in low-adhesion scenarios, enhancing safety through adaptive control, and demonstrating its feasibility for real-world applications. Notably, this strategy achieved significant energy recovery improvements, with regeneration gains of 11.6% for Driver 1, 10.4% for Driver 2, and 10.31% for Driver 3, calculated as averages across all tested trajectories. These results underline its effectiveness in optimizing braking performance on both dry and icy roads. Additionally, the approach highlights its adaptability and superior performance to traditional methods, particularly in minimizing false alerts and maximizing deceleration efficiency. The findings offer actionable recommendations for designers and engineers. Adaptive braking systems should prioritize dynamic adjustments to varying road and weather conditions, leveraging advanced sensors for real-time monitoring. In regions with frequent icy conditions, such as northern Europe or mountainous areas, these strategies deliver significant benefits in improving both safety and energy recovery. In contrast, for more stable climates, simplified braking systems can provide cost-effective solutions while maintaining essential functionality. Furthermore, dynamic programming and predictive models should be explored to refine braking systems, tailoring them to specific driving behaviors and environmental scenarios. While this study represents a significant step forward in regenerative braking strategies, future research could further enhance the model by incorporating additional weather factors. For example, integrating wind speed as a dynamic parameter could offer deeper insights into its effects on energy recovery and vehicle stability. Wind resistance, particularly at higher speeds, significantly impacts energy consumption, and real-time wind speed estimation could help optimize braking strategies to mitigate these effects. Additionally, accounting for the influence of ambient temperature on battery performance and energy regeneration efficiency would provide a more comprehensive model. Since batteries exhibit reduced efficiency in extreme cold or heat, temperature-dependent adjustments to regeneration settings could maximize energy recovery while protecting battery health. In conclusion, this study offers an adaptive, efficient, and practical solution to regenerative braking, tailored to diverse climatic and road conditions. It provides valuable insights for the development of next-generation electric vehicles, advancing energy optimization, safety, and system reliability in a variety of driving scenarios.]”
|
Reviewer 3 Report
Comments and Suggestions for Authors
This article proposes a Weather-Adaptive Regenerative Braking Strategy (WARBS) system, which leverages onboard sensors and data processing capabilities to enhance the energy efficiency of regenerative braking across diverse weather conditions, while minimizing unnecessary alerts. Here are some comments:
1. Figure 1 mentioned in Section 2.2 does not appear.
2. In Section 2, there are too few experimental samples related to the impact of weather on driving style. Can they be verified in relevant public datasets?
3. In Section 3.3.1, there is no in-depth explanation of why these features are selected as the features for driving style recognition, and there is a lack of relevant literature support. For the selection and basis of driving style recognition features, you can refer to the following literatures: IEEE Transactions on Intelligent Vehicles, 2023, 8(11): 4599-4612 and IEEE Transactions on Intelligent Transportation Systems, 2017, 19(3): 666-676.
4. In Section 3.3.3, the review of existing driving style recognition models can refer to a reference of IEEE Transactions on Intelligent Vehicles, 2023, 8(11): 4599-4612. And the review of driving style recognition models should be simplified or put into the literature review section. The reasons for choosing Decision Trees, SVM, and ANFIS as driving style recognition model should be explained.
5. In Section 4.1.1, the driving style recognition results need to be explained, such as what kind of driving characteristics drivers 1, 2, and 3 exhibit respectively.
Overall, the article should be major revised before being accepted.
Author Response
Comments 1: [Figure 1 mentioned in Section 2.2 does not appear.] |
|
Response 1: Thank you for highlighting this oversight. We agree with your comment. Figure 1 was indeed missing from the original submission. We have now included Figure 1 in the revised manuscript. The figure has been added on Page [7], in the section "[2.2. Reactions to braking distance]", immediately following Line [286]. This addition provides the necessary context and visual support for the discussion in this section. The updated figure has also been labeled and referenced appropriately in the text to ensure clarity. Please see the highlighted changes in the revised version for your review.]”
|
Comments 2: [In Section 2, there are too few experimental samples related to the impact of weather on driving style. Can they be verified in relevant public datasets?] |
|
Response 2: Thank you for highlighting this important point. We agree with your observation and have revised the manuscript to address this concern. We have incorporated data from the SHRP 2 Naturalistic Driving Study (NDS), a comprehensive dataset that includes real-world driving behavior under various road and weather conditions. This dataset allows for the analysis of how factors such as rain, snow, and other weather variables influence driving styles, including speed, acceleration, and braking behavior. In Section 2.1, we expanded the discussion to include the use of SHRP 2 NDS data for validating our findings. The relevant modifications can be found on Page [5], Paragraph [1], Line [206].]” “[To enhance the analysis of weather-related impacts on driving styles, we have combined experimental data with publicly available data from the SHRP 2 Naturalistic Driving Study (NDS) to extend our database. This comprehensive dataset captures real-world driving behavior under a variety of weather conditions, including rain and snow, complementing the controlled experimental data collected in our study. This integration ensures compatibility by harmonizing key variables, such as speed, acceleration, and braking patterns, across both data sources. By leveraging the strengths of both experimental precision and the representativeness of public data, we validate our findings and improve the study's robustness. The inclusion of diverse datasets ensures a more reliable evaluation of the model’s performance across a wide range of environmental scenarios, highlighting its adaptability and effectiveness.]”
|
Comments 3: [In Section 3.3.1, there is no in-depth explanation of why these features are selected as the features for driving style recognition, and there is a lack of relevant literature support. For the selection and basis of driving style recognition features, you can refer to the following literature: IEEE Transactions on Intelligent Vehicles, 2023, 8(11): 4599-4612 and IEEE Transactions on Intelligent Transportation Systems, 2017, 19(3): 666-676.] |
|
Response 3: Thank you for pointing this out. We agree with your comment and have revised the manuscript to provide an in-depth explanation of why these features were selected for driving style recognition, incorporating relevant literature to support our approach. We have expanded Section 3.3.1 to include a detailed rationale for selecting the features used in driving style recognition. Specifically, we have cited the suggested references, IEEE Transactions on Intelligent Vehicles (2023, 8(11): 4599-4612) and IEEE Transactions on Intelligent Transportation Systems (2017, 19(3): 666-676), as well as additional literature to substantiate the feature selection process. These updates can be found on Page [10], Section 3.3.1, Paragraph [1], Lines [372], and are marked in red in the revised manuscript]” “[We selected these features based on their demonstrated relevance in previous studies. For instance, [48] highlights their essential role in identifying driving behaviors, whether aggressive or cautious. Similarly, [49] emphasizes their effectiveness in recognizing driving patterns under different road conditions. In our previous research [32] , we also explored this issue by applying Principal Component Analysis (PCA) to reduce the dimensionality of the dataset used in the prediction model. The correlations between the principal components and the original variables were presented in Table 2, revealing no correlation between the principal components themselves, thereby ensuring their statistical independence. The first principal component was found to be strongly correlated with six key variables: battery current, battery voltage, battery power, motor torque, and, to a lesser extent, the distance to stop and speed scores. These results indicate that these six criteria vary together: an increase in one typically leads to an increase in the others. Furthermore, this component is particularly influenced by battery-related variables. The second principal component, on the other hand, is primarily correlated with rotational speed and its equivalent speed. It also shows a strong correlation with road characteristics obtained via GPS data. These analyses indicate that the following features are discriminative and relevant for our study. They are particularly effective in capturing driver responses to environmental factors, such as icy or wet roads, which directly influence driving style recognition.]”
|
Comments 4: [In Section 3.3.3, the review of existing driving style recognition models can refer to a reference of IEEE Transactions on Intelligent Vehicles, 2023, 8(11): 4599-4612. And the review of driving style recognition models should be simplified or put into the literature review section. The reasons for choosing Decision Trees, SVM, and ANFIS as driving style recognition model should be explained.] |
|
Response 4: Thank you for pointing this out. We agree with your comment and have revised the manuscript to provide an in-depth explanation of why these features were selected for driving style recognition, incorporating relevant literature to support our approach. The review of existing driving style recognition models has been moved from Section 3.3.3 to the introduction to improve organization and provide a clearer context at the beginning of the manuscript. In Section 3.3, we have added a detailed explanation of the reasons why Decision Trees, SVM, and ANFIS were selected as the primary models for driving style recognition and cited the review of driving style recognition methods from short-term and long-term perspectives. These updates can be found on Page [11], Section 3.3.3, Paragraph [1], Lines [409], and are marked in red in the revised manuscript. ]” “[The driving style recognition models evaluated in this study include Decision Trees, SVM, and ANFIS, each selected for its unique strengths and proven effectiveness. Decision Trees stand out for their simplicity and high classification accuracy, particularly when applied to time-series data. SVM excels in managing non-linear patterns and high-dimensional datasets, ensuring reliable differentiation of driving styles. Meanwhile, ANFIS combines the advantages of fuzzy logic and neural networks, allowing for in-depth and nuanced analysis of driving behaviors. These models were chosen based on their complementary strengths and their demonstrated success in prior studies, including research published in [ 48 ]. Our research goes beyond a straightforward comparison of Decision Trees, SVM, and ANFIS by focusing on enhancing their accuracy through systematic optimization of structural and training parameters. To achieve this, we employ advanced feature extraction and selection techniques, ensuring that the models are equipped to interpret complex datasets with greater precision. These methodologies are central to improving the robustness and effectiveness of the models, enabling them to capture intricate patterns and deliver reliable driving style recognition across diverse scenarios.]” [48] H. Chu, H. Zhuang, W. Wang, X. Na, L. Guo, J. Zhang, B. Gao, H. Chen, A review of driving style recognition methods from short-term and long-term perspectives, IEEE Transactions on Intelligent Vehicles 8(11) (2023) 4599-4612. [49] C.M. Martinez, M. Heucke, F.-Y. Wang, B. Gao, D. Cao, Driving style recognition for intelligent vehicle control and advanced driver assistance: A survey, IEEE Transactions on Intelligent Transportation Systems 19(3) (2017) 666-676. |
Comments 5: [In Section 4.1.1, the driving style recognition results need to be explained, such as what kind of driving characteristics drivers 1, 2, and 3 exhibit respectively.] |
|
Response 5: Thank you for pointing this out. We agree with your comment and have revised the manuscript to address the explanation of the driving style recognition results in Section 4.1.1. We have added a detailed explanation of the driving characteristics exhibited by Drivers 1, 2, and 3. Specifically, we describe the unique driving behaviors of each driver, including their tendencies toward aggressive, cautious, or balanced driving styles. This analysis is based on key parameters such as speed, acceleration, braking patterns, and responsiveness to weather factors. In Section 4.1.1, Page [17], Paragraph [1], Lines [607], we have included a breakdown of how each driver’s behavior influenced the model's recognition results.]” “[The drivers are analyzed based on distinct characteristics, such as maximum speed, longitudinal acceleration, and jerk values (rate of change of acceleration). Driver 1 exhibits a dynamic driving style, particularly noticeable on dry roads, with higher maximum speeds and acceleration values compared to the other drivers. On icy roads, while speeds are reduced, the style remains relatively aggressive, with significant variations in acceleration and jerk parameters. Driver 2 adopts a moderate style, characterized by lower maximum speeds and more stable driving. The smooth speed adjustments and minimal variations in acceleration values reflect a cautious approach, especially in slippery conditions. Driver 3 stands out with a conservative driving style, featuring significantly lower maximum speeds and reduced jerk and acceleration values, demonstrating an adaptation to slippery road conditions.]”
|
Round 2
Reviewer 2 Report
Comments and Suggestions for Authors
The authors did a very good job. The authors were very responsible in correcting the article. Almost all sections of the article were corrected and my questions were answered. The article became significantly better.
Reviewer 3 Report
Comments and Suggestions for Authors
No further comment